# Adsorption Processing for the Removal of Toxic Hg(II) from Liquid Effluents: Advances in the 2019 Year

**Francisco J. Alguacil** and **Félix A. López** *

National Center for Metallurgical Research (CENIM), Spanish National Research Council (CSIC), Avda. Gregorio del Amo 8, 28040 Madrid, Spain; fjalgua@cenim.csic.es
* Correspondence: f.lopez@csic.es; Tel.: +34-9155-38900

**Abstract:** Mercury is a toxic metal, thus, it is an element which has more and more restrictions in its uses, but despite the above, the removal of this metal, from whatever the form in which it is encountered (zero valent metal, inorganic, or organic compounds), and from different sources, is of a widespread interest. In the case of Hg(II), or $Hg^{2+}$, the investigations about the treatment of Hg(II)-bearing liquid effluents (real or in most cases synthetic solutions) appear not to end, and from the various separation technologies, adsorption is the most popular among researchers. In this topic, and in the 2019 year, more than 100 publications had been devoted to this field: Hg(II)-removal-adsorption. This work examined all of them.

**Keywords:** mercury(II); adsorption; processing; liquid effluents

## 1. Introduction

From earlier times, and together with gold, silver, copper, iron and lead, mercury is one of the firsts metals used by man, and despite its past and recent usefulness, nowadays, mercury is considered one of the most hazardous metals against humans and aquatic environments, and this is because its ingesta, in whatever mercury form (zero valent metal, inorganic, and organic compounds), is responsible for a number of diseases and even death. It is evident, that the presence of mercury in the environment is caused by natural and anthropogenic causes, being the latter which probably most contributed to the presence of the metal in these environments. It is also evident that, besides all the limitations and bans in its uses, mercury is still producing in various countries. The 2019 mercury world mine production is distributed, from near a total of 4000 t, as: China 3500 t, Mexico (export) 240 t, Tajikistan 100 t, Peru (export) 40 t, Argentina 30 t, Kyrgyztan 20 t, Norway 20 t and other countries 20 t [1].

It is worth to note here that with respect to tolerances in mercury presence in waters, there are similarities and discrepancies between the various organizations, i.e., for the U.S. Environmental Protection AgencyUSEPA, mandatory discharge limit of 10 µg/L for wastewater and 1 µg/L for drinking water is the norm, whereas in the case of the World Health Organization WHO, the limits are fixed to 5 µg/L and 1 µg/L, respectively. European Union, under its normative 98/83/EC, fixed the maximum mercury content in drinking water as 0.001 mg/L.

In aqueous solutions, mercury can be found as Hg(II), or $Hg^{2+}$, forming a number of salts, e.g., $Hg(NO_3)_2$, and even it can be found forming anionic complexes, e.g., $HgCl_4^{2-}$. Particularly, the presence and removal of $Hg^{2+}$ from aquatic environments is one of the major topics of interest to worldwide scientists, and among the separation technologies used for this task: liquid-liquid extractions, membranes, ion-exchange, precipitation, adsorption is probably the most popular, and this is demonstrated by the number of Hg(II)-adsorption papers published in Journals every year and in comparison with these published using the others separation technologies, i.e., in 2019, more

than 100 papers were dedicated to adsorption against no more than 10 using all together the others technologies [2].

This work reviewed all this Hg(II)-adsorption papers published during 2019. They showed a variety of adsorbents for Hg(II), but many of these papers did not account for the desorption process and all of them did not consider what to do with the final Hg(II)-solution.

## 2. Hg(II) Removal Using Adsorption Methodologies

(Mn(BDC)(H$_2$O)$_2$) adsorbent (MN-metal-organic framework (MOF)) was synthesized by the use of a microwave technology [3]. Its adsorption characteristics were investigated against a series of heavy metals: Hg(II), Cd(II), and Pb(II). In the case of mercury(II), this adsorption reached 65%, a value which was lower than that found for Cd(II) (82%) and Pb(II) (71%). The adsorption of mercury(II) fitted well to the Langmuir isotherm:

$$\frac{1}{[\text{Hg}]_{a,e}} = \frac{1}{[\text{Hg}]_{aq,e}[\text{Hg}]_{a,m}k_L} + \frac{1}{[\text{Hg}]_{a,m}} \tag{1}$$

where [Hg]$_{a,e}$ and [Hg]$_{aq,e}$ were the mercury concentrations in the adsorbent and in the aqueous solution at the equilibrium, respectively, [Hg]$_{a,m}$ was the maximum mercury concentration in the adsorbent, and k$_L$ was the Langmuir constant. Adsorption kinetics fitted the pseudo-second-order kinetic equation:

$$\frac{t}{[\text{Hg}]_{a,t}} = \frac{1}{K_2[\text{Hg}]_{a,e}^2} + \frac{t}{[\text{Hg}]_{a,e}} \tag{2}$$

in this equation, [Hg]$_{a,t}$ and [Hg]$_{a,e}$ were the mercury concentrations in the adsorbent at an elapsed time and at the equilibrium, respectively, t was the elapsed time, and K$_2$ was the constant. Apparently, desorption was performed using acetone. This adsorbent seemed to lost their adsorption properties since after the fifth cycle the percentages of adsorption were 51% for Hg(II) and 68% and 55% for Cd(II) and Pb(II), respectively.

A diacid Schiff base ligand was synthesized via the reaction of 2-hydroxybenzaldehyde with 5-aminoisophthalic acid; in a further reaction, Schiff base nanostructure aromatic polyamide was prepared by the reaction of 4,4'-diaminodiphenyl ether and the synthesized diacid in the presence of molten tetrabutylammonium bromide (TBAB)/triphenyl phosphite (TPP) as the green reaction medium [4]. Then, the metal-polymer complex was synthesized from polyamide ligand and manganese(III) acetate dihydrate salt. The as-synthesized nanostructure was investigated in the removal of Hg(II) (and Cd(II)) from aqueous solutions. Whereas the equilibrium was achieved in 30 min at pH 7, Hg(II) adsorption fitted the Langmuir isotherm (maximum Hg(II) adsorption capacity of 0.06 mmol/g) and pseudo-second-order kinetic model. The adsorption process was exothermic. Hg(II) bonded to the oxygen of the carbonyl groups and nitrogen of the NH groups of the adsorbent. Elution was performed by the use of 0.01 M acetic acid, and apparently it was reused up to three cycles without loss in the adsorption capacity.

A series of sulfone-modified chitosan derivatives had been synthesized and applied to the adsorption of Hg(II) [5]. Among them, the 5-derivative showed the best adsorption efficiency at pH 2, with an adsorption capacity for mercury(II) of 0.61 mmol/g, being the adsorption isotherm in agreement with the Langmuir adsorption isotherm, written as:

$$\frac{[\text{Hg}]_{aq,e}}{[\text{Hg}]_{a,e}} = \frac{[\text{Hg}]_{aq,e}}{[\text{Hg}]_{m,e}} + \frac{1}{[\text{Hg}]_{a,m}k_L} \tag{3}$$

No data about the desorption process were given in the publication.

A nanocomposite include Schiff base polymer and magnetic Fe$_3$O$_4$ nanoparticles was synthesized and characterized. The polymer (MPOP) was used as an adsorbent for the removal of Hg(II)

(and Cd(II)) from aqueous solution [6]. The adsorption kinetic results revealed that Hg(II) fitted the pseudo-second-order adsorption kinetic model and the Langmuir isotherm, with a maximum adsorption capacity of 1.42 mmol/g. The adsorption process was demonstrated to be spontaneous and endothermic. Elution was carried out with 0.1 M HCl solutions, and after six cycles of adsorption-desorption there was not apparent loss in the adsorption capacity of the adsorbent. Analysis of the experimental data, indicated that Hg(II) (and Cd(II)) adsorption occurred via coordination-covalent bonds with the phenolic and azomethinegroups of the MPOP.

A new resin was synthesized by cyclo-copolymerization of dialylammonioethanoate and maleic acid, and in the presence of a cross-linker [7]. Due to the presence of the chelating glycine residues was, the new resin can be used in the removal of Hg(II), with a maximum metal uptake of 1.31 mmol/g. The simultaneous removal of methylene blue and Hg(II) from industrial wastewater it is also feasible. No data about Hg(II) desorption were given in the manuscript.

Carboxylic-functionalized multi-walled carbon nanotubes (MWCT-COOH) were subjected to a functionalization process with 3-amino-5-phenylpyrazole (MWCNTs-f) [8], and characterized by the usual techniques. Both materials were used to investigate the adsorption of Cd(II), Hg(II), and As(III) from aqueous solutions. Several experimental parameters were investigated on the metal adsorption process: the influence of pH, adsorbent dosage, and initial metals concentration, furthermore, central composite design (CCD) was also used in the study. The model indicated that the adsorption process strongly depends on the pH of the aqueous solution, being these adsorptions well described by the pseudo-second-order kinetic model and the Langmuir isotherm. The adsorption of Hg(II) (and also Cd(II) and As(III)) were spontaneous and endothermic, whereas the competitive adsorption process was slightly lower than the non-competitive results. By the use of MWCNTs-f, similar affinity order: As(III) > Cd(II) > Hg(II), resulted from both non-competitive and competitive processes. Desorption investigations showed the regeneration ability of both materials.

Bovine bone wastes were used as starting materials to prepare bio-apatite adsorbents using thermal treatments with a direct flame and annealing at 500–1100 °C [9]. As being usual, these materials were characterized and investigated as Hg(II) adsorbents. A Central composite design (CCD)—Response surface methodology (RSM) design was used to optimize and analyze experimental variables such as: initial mercury concentration (10–100 mg/L), aqueous pH (2–9), adsorbent dosage (0.1–0.5 g), temperature (20–60 °C), and reaction time (15–120 min). Experimental results showed that Hg(II) uptake depended on the temperature from which the material was annealed, and material type: 500 °C > 600 °C > 800 °C > 1100 °C > direct flame > starting material. Hg(II) were eliminated from the solution by dissolution-precipitation (low pH values) or ion-exchange (high pH values) reactions are the two dominant mechanisms for the removal of Hg(II) ions at low and high pH values, respectively. The CCD-RSM predicted maximum mercury adsorption of 99.99% under the optimal conditions of 51 mg/L, 0.44 g, 6.5, 67.5 min, and 50 °C for initial mercury concentration, adsorbent mass, pH, contact time, and temperature, respectively. The findings of the present study revealed that the bio-apatite based materials, particularly BB500, are suitable and versatile adsorbents for the treatment of mercury-containing wastewater. No data about metal desorption were given in the manuscript.

A process based in the functionalized/reduction with polyethylenimine of graphene oxide embedded calcium alginate (GOCA) was developed to increase the adsorption capacity of GOCA towards heavy metal ions [10]. Thus, the adsorption of Hg (II), among other metals, from aqueous solutions was investigated under various experimental conditions, having the functionalized beads a high adsorption capacity compared to the non-functionalized beads (Table 1).

Maximum Hg(II) uptake was of 1.86 mmol/g, following the adsorption process the pseudo-second-order kinetic model and the Langmuir isotherm. Elution was performed using a solution of pH 5 (HCl medium) and 0.2 g/L calcium chloride. There was a continuous decrease if Hg(II) adsorption from near 99% (1st cycle) to near 75% (5th cycle).

**Table 1.** Percentage of Hg(II) adsorption using various types of alginate beads.

| Adsorbent | %Hg (II) Removal |
|---|---|
| Calcium alginate (CA) | 45 |
| Graphene oxide embedded calcium alginate (GOCA) | 65 |
| Functionalized GOCA | 95 |

Adsorbent: 4% wt SA and 5 mg GO/mL water. Aqueous solution: 0.05 g/L Hg(II). Temperature: 25 °C. Time: 12 h.

A new adsorbent, a type of nanoporous silica material, was developed to remove mercury(II) from water [11]. Three variables: pH value of the solution, contact time, and adsorbent dosage were experimentally considered, with best Hg(II) uptake at pH 4.5, reaction time 25 min, and adsorbent dosage of 55 mg/L. Maximum Hg(II) adsorption was 0.57 mmol/g, with the data fitted to the Langmuir isotherm. There was not interference of ions ($Na^+$, $K^+$, Mg(II), Pb(II), Cd(II), Ni(II) but $Cl^-$) on Hg(II) uptake. No data were included about which acid was used in the desorption step, but Hg(II) uptake slightly decreased from near quantitative adsorption to 94% from the 1st to the 5th cycle. The adsorbent was used to remove Hg(II) from a real effluent from Bandar Emam Petrochemical Unite. The effluent had a pH of 6.5 and the mercury(II) removal efficiency was of 57%.

The adsorption of Hg(II) on kaolinite(001) surface in an aqueous environment was explored using density functional theory (DFT) calculations [12]. The preferred coordination number of 5 was maintained with complexes containing 3-4 aqua ligands in the stabilized structures, whereas energy calculations showed that Hg(II) had strong interactions on partially deprotonated surface attributable to the existing chemical bonding. With respect to the bonding nature, it was found that Hg-O-s (surface-O) bonds had covalent characteristic, ascribing to the Hg-5d and O-s-2p orbital overlap, accompanying with charge transfer from the substrate to adsorbate.

A novel thiol-modified magnetic activated carbon adsorbent of $NiFe_2O_4$-powdered activated carbon (PAC)-SH was synthesized via a hydrothermal method [13]. The as-prepared adsorbent removed Hg(II) from aqueous solution with a maximum uptake, at pH 7, of 1.49, and 1.82 mmol/g based on the experimental and Langmuir fitting, respectively. Ion exchange and electrostatic attraction were the main adsorption factors in an exothermic and spontaneous process. Desorption was performed using 0.1 M HCl solutions; after five consecutive adsorption-desorption cycles, the adsorption capacity was reduced in an 18% (1.23 mmol/g) with respect to that obtained in the first cycle.

A thermal precipitation method was used to prepare nanosheet CuSbOS bimetal oxysulfide catalysts, and characterized by the usual technologies [14]. CuSbOS reduction activities were investigated by the reduction of Hg(II) (also Cr(VI) and Pb(II)) aqueous solution and organic dyes such as methyl orange (MO), rhodamine-B (RhB), and methylene blue (MB). The results showed that CuSbOS were excellent in reduction activity for Hg(II) and the other metals, without adding any other agents, and for MO, MB, and RhB but with the addition of $NaBH_4$.

Different techniques had been used to characterize a tubular bimetal oxysulfide CuMgOS catalyst, which was prepared at a temperature of 95 °C [15]. The CuMgOS reduction power was investigated through the reduction of Hg(II) in solution, being this metal completely removed, from a 0.05 g/L Hg(II)-bearing solution, with 0.2 g/L of the adsorbent and 4 min of reaction. The work also gave data about the removal of Cr(VI), Pb(II), and the organic dyes of rhodamine-B (RhB), methyl orange (MO), and methylene blue (MB).

Two new thiol-/thioether-functionalized porous organic polymers were prepared, and characterized by the usual methods, in order to investigate their abilities in the removal of Hg(II) from water which also contained aromatic pollutants [16]. Hg(II) was effectively removed at pH values of 3–4, presenting these new adsorbents a maximum Hg(II) adsorption capacity of 0.90 mmol/g (1.78–2.25 mmol/g for aromatic pollutants).The adsorbents also showed high adsorption selectivity for Hg(II) in the presence of other metals in the solution. Desorption was performed by the use of 6 M HCl solution, and there was a decrease on Hg(II) removal efficiency after three cycles (from 95% to 86%). Furthermore, the two adsorbents were effective in the simultaneous removal of Hg(II) and

aromatic pollutants in simulated sewage, with removal efficiencies higher than 88% for Hg(II) (93% for the aromatic pollutants) from a sewage containing 10 μg/mL of each one using a dosage of 1 g/L of the adsorbents.

A fluorescent supramolecular polymer has been constructed from a newly designed [2] biphenyl-extended pillararene [6] equipped with two thymine sites as arms (H), and a tetraphenylethylene (TPE)-bridged bis(quaternary ammonium) guest (G) with aggregation-induced emission (AIE) property [17]. Moreover, supramolecular assembly-induced emission enhancement (SAIEE) switched on, upon addition of Hg(II) into the above-mentioned supramolecular polymer system, to generate spherical-like supramolecular nanoparticles. This supramolecular polymer, with integrated modalities, had been used for real-time detection and removal of Hg(II) and recycled with $Na_2S$. The system combined the rigid and spacious cavity of novel extended-pillarene host with the AIE characteristics of TPE-based guest. 1,8-(3,6-dithiaoctyl)-4-polyvinyl benzenesulphonate (dpvbs) resin was used as an effective adsorbent for the removal of Hg(II) from aqueous solutions [18]. Spectral data indicated that Hg(II) was adsorbed onto dpvbs through sulfur and oxygen atoms, and in the form of mercury(II) nitrate. The maximum capacity of the resin was found to be of 2.17 mmol/g, with data fitted to the pseudo-second-order kinetic model and Freundlich isotherm:

$$\ln[Hg]_{a,e} = \ln K_f + \frac{1}{n} \ln[Hg]_{aq,e} \tag{4}$$

It was proposed that Hg(II) was uptaken onto the resin by physisorption. No desorption information was given in the manuscript.

Novel water treatment residual nanoparticles (nWTR), produced by precision milling, were evaluated to the removal of Hg(II) from water [19]. The nWTR had a Hg(II) removal efficiency of 93% within 15 min, with Langmuir and Temkin isotherm models describing the adsorption equilibrium, the former represented by:

$$[Hg]_{a,e} = \frac{RT}{b} \ln K_T + \frac{RT}{b} \ln[Hg]_{aq,e} \tag{5}$$

where b was the Temkin constant, which is related to the heat of adsorption, and $K_T$ was Temkin isotherm constant. Hg(II) adsorption kinetics data were best described by the power function model:

$$\log[Hg]_{a,t} = \log a + b \log t \tag{6}$$

where a and b were parameters associated with the model and derived from the plot of experimental data. It was suggested that the hydroxyl groups were the surface active sites for Hg(II) binding to nWTR surfaces. Hg(II) removal decreased as the solution pH increased its alkalinity (from 3 to 11).

Gum Arabic (*Acacia gum* (GA)) dialdehyde phenylthiosemicarbazone (GAD-PTSC) chelating resin was prepared through the oxidation of GA followed by a reaction with 4-phenylthiosemicarbazide (PTSC), and used in the removal of Hg(II) from solutions [20]. The percentage of Hg(II) removal was 97% at pH 5.5, and the maximum Hg(II) uptake was found to be 120 mg/g at the same aqueous pH value. The experimental results fitted the second-order kinetic model, which indicated a chemical adsorption process. The Langmuir isotherm model also fitted well to the experimental results.

Different experimental conditions were used to optimize the Hg(II) adsorption by Eucalyptus globulus bark [21]. Experimental conditions were: pH (4, 6.5, and 9.), salinity (0.15 and 30), and biosorbent dosage (0.2, 0.5, and 0.8 $g/dm^3$), and initial Hg(II) concentration of 50 $m\mu g/dm^{-3}$. Best conditions for Hg(II) adsorption (81%) were: pH value of 6, no salinity, and a biosorbent dosage of 0.55 $g/dm^{-3}$. Kinetics obeyed the pseudo-second-order equation, and the increase of the ionic strength influence Hg(II) adsorption decreasing it. No elution data was given in the work.

A novel adsorbent thiol functionalized magnetic carbon nanotubes (CNTs-SH@$Fe_3O_4$) was synthesized and used to adsorb Hg(II) from water [22]. Batch experiments showed that this adsorbent had a high Hg(II) adsorption efficiency (>98%) over wide pH range (3–11), being Hg(II) adsorption exothermic and selective in the presence of Cu(II), Mg(II), or Zn(II) in the solution. Hg(II) adsorption

kinetics was best described by the pseudo-second-order kinetic model and the Freundlich isotherm, with a calculated maximum uptake of 0.86 mmol/g. Surface adsorption, complex adsorption, and reduction adsorption were all contributed to the removal of Hg(II) using this adsorbent.

Desorption step was done using a 1 M $HNO_3$ solution and under sonication, then washed with ethanol and dried (60 °C). After five cycles, the removal of Hg(II) decreased from 94% to 80% (Figure 1).

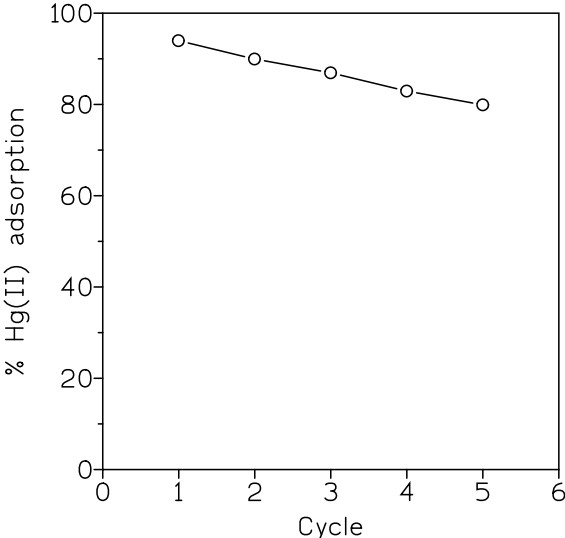

**Figure 1.** Lost of Hg(II) adsorption onto the adsorbent after various adsorption-desorption cycles.

A composite platform was synthesized and used for Hg(II) optical sensing and magnetic removal [23]. There was a core-shell structure in the composite platform, using superamagnetic $Fe_3O_4$ as core, silica molecular sieve MCM-41 as shell. As chemosensors, two rhodamine derivatives were synthesized and covalently grafted into MCM-41 layer. These composite samples showed emission turn on effect towards Hg(II) with good selectivity, and linear sensing curves were obtained. Maximum Hg(II) adsorption was determined as high as 0.02 mmol/g, and ethylendiaminetetraacetic acid (EDTA) was used to remove Hg(II) from the loaded material. The adsorption properties were repeated at least for four cycles.

An MOFs adsorbent was prepared by post-functionalization of UiO-66-$NH_2$ with 2,5-dimercapto-1,3,4-thiadiazole and used to remove the Hg(II) in water [24]. At a pH value of 3, the maximum adsorption capacity was 3.34 mmol/g. Adsorption kinetic was fitted to the pseudo-second-order model, whereas isotherms fitted to the Langmuir model as well as the Dubinin-Radushkevich model:

$$\ln[Hg]_{a,e} = \ln[Hg]_{a,m} - K\varepsilon^2 \tag{7}$$

where K was a constant related to the adsorption energy and $\varepsilon$ was the Polanyi potential, expressed as:

$$\varepsilon = RT\left(1 + \frac{1}{[Hg]_{aq,e}}\right) \tag{8}$$

Metal uptake was geared to monolayer and chemisorption, being the removal rate directly proportional to the square of mercury(II) concentration. Metal uptake was related to a complexation reaction between Hg(II) and thiol/nitrogen-containing groups, being Hg(II) adsorbed selectively in the presence of Zn(II), Co(IV), Ni(II), Cd(II), Mg(II), Fe(III), Ca(II), and Cu(II) in the solution. As eluent, 10% thiourea was used, and after ten cycles, the adsorption decreased from near 99% to 85%.

Magnetic $Fe_3O_4$-encapsulated $C_3N_3S_3$ polymer/reduced graphene oxide composite (rGO-poly($C_3N_3S_3$/$Fe_3O_4$)) was synthesized to Hg(II) (and Pb(II)) from aqueous solutions [25]. The composite exhibited two-dimensional (2D) nanosheet structure, in which $Fe_3O_4$ nanoparticles or clusters

are encapsulated between the layers of rGO-poly($C_3N_3S_3$) matrix, preventing composite aggregation and nanoparticle detachment. The adsorption kinetics data were well fitted by the pseudo-second-order model, being the equilibrium data represented by the Langmuir isotherm. Maximum Hg(II) uptake was of 1.99 mmol/g (1.30 mmol/g in the case of Pb(II)). Hg(II) was preferentially bonded to sulphur functional groups.

A new material (MAP) was manufactured by inducing the polymerization of m-aminothiophenol onto MCM-41 (Mobil Composition of Matter) surface [26]. After 15 min of reaction, Hg(II) was removed from the solution with yields of 97% (1.21 mmol/g), being the system competitive against the presence of other metals in solution Na(I), Mg(II), K(I), Al(III), Ca(II). The adsorption of mercury(II) fitted the pseudo-second-order model, and the adsorption, whereas equilibrium data responded well to the Langmuir model. A mixed solution of 0.2 M nitric acid and 0.1 M thiourea was used as eluent, after five cycles, the adsorbent lost its adsorption capacity, decreased (in a continuous form) until 0.95 mmol/g, the authors explained this decrease due to associated adsorbent material loss during the repeating experience The Hg(II)-loaded adsorbent was used as a catalyst in the reaction of phenylacetylene to acetophenone, reaching yields of 97%.

In the next investigation [27], functionalized magnetic mesoporous silica/organic polymers nanocomposite (MMSP) was fabricated by the grafted poly(m-aminothiophenol), which embedded the aminated magnetic mesoporous silica nanocomposite based on $Fe_3O_4$ magnetic core, which was shelled by mesoporous silica and further modified by (3-aminopropyl) triethoxysilane. The results indicated that the mercury(II) removal percentage of the MMSP reached 98% (1.22 mmol/g) within 10 min at pH 4.0, having the coexisting ions (the same than in the previous reference) have no significant effect on the selective Hg(II) removal from the solution. Hg(II) uptake was spontaneous and exothermic, and the adsorption fitted to the pseudo-second-order model and Langmuir isotherm. Using the same eluant as in the above reference and similarly to it, the system lost its adsorption capacity in a continuous form from the first to the fifth cycle. After mercury(II) adsorption, the material was an efficient catalyst for transformation of phenylacetylene to acetophenone, with yields of 98%.

Magnetic $Fe_3O_4$ functionalized by guanidine acetic acid ($Fe_3O_4$@GAA), as a nano-adsorbent (20–30 nm diameter), was used in the removal of Hg (II), Zn (II), Cu (II), and Ag (I), from aqueous solutions [28]. The best adsorptive conditions were at pH 7, nano-adsorbent amount of 14 mg, and reaction time of 10 min. Next ions: K(I), Na(I), Li(I), Mg(II), Fe(II), Ca(II), Ba(II), Mn(I), Sn(II), Ni(II), Co(II), $NO_3^-$, $Cl^-$, and $SO_4^{2-}$ did not influence the adsorption of Hg(II) onto the adsorbent. As eluent, a solution of pH 3 (nitric acid medium) was used, and after twenty consecutive cycles, a removal efficiency exceeding 95% was maintained. Black phosphorus monolayer (BPML) material was investigated in order to remove cadmium, lead, mercury, and arsenic(III) from solutions [29]. The coordination of metal ion varied from two to four. Both expanding and contracting distortions were observed. The bandgap of BPML was enlarged by 0.12–1.10 eV, with the smallest and largest changes for Pb (II) and Hg (II), respectively. After decoration, all decorated BPML nanosheets kept their p-type semiconducting property. The considerably high adsorption energies followed the sequence Pb(II) < Cd(II) < Hg(II) << As(III), in full correspondence with the remarkable charge transfers (0.440–1.236 eV). The localized orbital locator (LOL) profiles revealed the nature of the interactions. The published paper lacked information about (i) conditions for adsorption, (ii) desorption.

The next reference [30] provided an insight on environmental application, aimed to the removal of mercury, of a variety of iron-based adsorbents: zero valent iron, iron oxides ($\gamma$-$Fe_2O_3$, $Fe_3O_4$, $\alpha$-$Fe_2O_3$), iron sulfides (FeS, $FeS_2$), as well as hydrous ferric oxides, i.e., $\delta$-FeOOH, and Fe-Mn binary oxides. The review included the use of nanomaterials, functionalized nanomaterials, and supported iron-based materials.

Activated carbon, from mixed recyclable waste, and modified by phosphonium-based ionic liquid (IL-ACMRW) was prepared and evaluated for Hg(II) removal from solutions [31]. The activated carbon used in this study was originated from cardboard, papers, and palm wastes, and mixed with trihexyl(tetradecyl)phosphonium bis2,4,4-(trimethylpentyl)phosphinate ionic liquid (commercial name

Cyphos IL104). The investigation showed that the uptake was 124 mg/g at pH 4 and after 90 min. The adsorption of Hg(II) was found to follow the pseudo-first and pseudo-second-order kinetics models and Freundlich isotherm, thus, suggesting multilayer adsorption. The published manuscript did not include data about the desorption process. Two Zn-based metal-organic frameworks (MOFs), named TMU-31 and TMU-32, were synthesized by a solid-state method, named mechanosynthesis, and were highly decorated with urea functional groups [32]. They were used in the adsorption of Hg$^{2+}$ (and Pb$^{2+}$) from wastewater. The adsorption capacity for Hg(II) was 2.37 mmol/g in less than 15 min (4.39 mmol/g for Pb(II) but in less than 5 min). As many adsorbents mentioned here, the adsorption isotherm for both metals responded well to the Langmuir model, and also fitted to the pseudo-second-order kinetic model.

Magnetic chitosan microparticles were functionalized by grafting a new hydrazide derivative to produce HAHZ-MG-CH [33], which presented a high nitrogen content (10.9 mmol/g). Mercury (II) maximum sorption capacity at pH 5 was of 1.79 mmol/g (also data for U and Cd were included in the work). Adsorption isotherm was fitted to the Langmuir model. In acidic solutions, this adsorbent had a marked preference for Hg(II), however, at mild pH values, uranyl species are preferentially adsorbed. In multielemental solutions, the adsorbent has a lower affinity towards Cd(II). Equilibrium was attained after 60 min, and the data fitted well to the pseudo-first-order rate equation:

$$\log\left([Hg]_{a,e} - [Hg]_{a,t}\right) = \log[Hg]_{a,e} - \frac{k_1}{2.3}t \tag{9}$$

$k_1$ being the rate constant, and also to the pseudo-second-order equation. A 0.5 M HCl solution was used to remove Hg(II) loaded onto the adsorbent with yields exceeding 97% in the 1st cycle, though there was a 8% and 6% loss in adsorption and desorption capacities, respectively, after 5 cycles.

A micro-nano composite material was prepared via the co-precipitation method [34]. Oyster shell (OS), Fe$_3$O$_4$ nanoparticles, and humic acid (HA) were used as the raw materials. The effects of pH (3–7), initial solution concentration (2.5–30 mg/L), and contact time (0–5 h) on the adsorption of Hg(II) ions were investigated. The experimental data were well fitted to the Langmuir isotherm, and followed the pseudo-second-order reaction model. Maximum Hg(II) uptake was 142 mg/g at pH 5. No data about the removal of Hg(II) from Hg(II)-loaded adsorbent were given in the published manuscript.

Sulfur activated carbon (SAC) was used to remove Hg(II) from actual lime-based wet flue gas desulfurization (WFGD) wastewater [35]. The increase of the pH from 4 to 7 resulted in a 22% Hg(II) decrease of the adsorption capacity. The explanation for these results appeared to be that a low pH values, the next reaction occurred:

$$HgCl_2 + S - C \rightarrow HgS - C + 2Cl^- \tag{10}$$

whereas at higher pH values, Hg$^0$ was formed and it was re-emitted into the gaseous phase. Equilibrium data fitted the Freundlich isotherm, whereas both pseudo-second order and Elovich models described the chemisorption behavior of Hg(II) onto SAC, with the Elovich model represented by:

$$[Hg]_{a,t} = a + 2.3b \log t \tag{11}$$

again, a and b were model parameters estimated by the corresponding plot of the data. The adsorption process was demonstrated to be spontaneous and exothermic. No data were given about desorption step.

A sulfur-doped magnetic amide-linked organic polymer (S-MAOP), where AOP was chemically anchored on the surface of NH$_2$-functionalized magnetic nanoparticles, was synthesized and used in the removal of Hg(II) from solutions [36]. Maximum adsorption was obtained at pH 3, and the presence of coexisting ions in the aqueous solution decreased Hg(II) adsorption, this effect was more notorious in the case of CaCl$_2$, and as a general trend the effect increased as the concentration of the

salt (NaCl, CaCl$_2$, KCl, NaNO$_3$, and Na$_2$SO$_4$) in the solution increased from zero to 100 mmol/L. No desorption data included in the work.

A strategy for Hg(II) removal was developed by incubating algae cells in high-phosphate cultures, the as-obtained phosphorous-rich biomass was a used to adsorb Hg(II), and the Hg(II)-loaded biomass was charred to prevent leaching of phosphate and to immobilize the metal [37]. Algae surface modification was done using *Scenedesmus obtusus XJ-15*, which were put into contact with various phosphorous concentrations, 80 mg/L phosphorous solutions allowed to obtain the highest site concentrations of surface phosphoryl functional groups, which enhanced Hg(II) adsorption (Table 2).

**Table 2.** Effect of phosphorous concentration in the medium on Hg(II) adsorption.

| Phosphorous, mg/L | Hg(II) Adsorption, mmol/g |
|---|---|
| 0 | 0.29 |
| 20 | 0.42 |
| 40 | 0.39 |
| 80 | 0.47 |
| 110 | 0.41 |
| 120 | 0.43 |

This B-80 biomass (algal cells cultivated in BG-11 medium containing phosphorous concentration of 80 mg/L) had the highest adsorption uptake of 0.475 mmol/g at pH 5, with the results fitted to the Langmuir isotherm model. Charred process was done under various temperatures, having the product processed at 300 °C the lowest Hg(II) leaching rate and without phosphorous released to the solution. A biomass (CSS) was used as Hg(II) adsorbent [38]. The adsorbent material was prepared by co-heating sublimed sulfur and raw corn stalk at 450 °C and 15 min, resulting in a product with 7.4% (in weight) sulfur. The maximum Hg(II) uptakes were of 1.34 and 0.59 mmol/g for CSS and BCS, respectively. Experimental results showed that in CSS, there was formation of irreversible Hg–S and Hg–O bonds. Desorption data were not included.

Several variables were investigated in the adsorption of Hg(II) by low-rank Pakistani coal (LPC) [39]. Maximum Hg(II) adsorption was found to be $1 \times 10^{-4}$ M nitric acid solutions, using a 38 g/L adsorbent dosage and 10 min of reaction time. Metal uptake decreased with the increase of the acids used in the investigation (nitric, hydrochloric, etc.), being the Hg(II) loading onto the coal dominated by an intra particle diffusion process and the pseudo-second order kinetic model, whereas the adsorption data obeyed the Langmuir, Freundlich and Dubinin-Radushkevich isotherm plots. Mercury(II) uptake was increased with the increase of the temperature (10–60 °C). The manuscript lacked desorption data.

Pectin (PC)-g-[methacrylic acid (MAA)-co-3-(N-isopropylacrylamido)-2-methylpropanoic acid (NIPAMPA)-co-N-isopropylacrylamide (NIPA)] (PC-g-TP) was used in the adsorption of Hg(II) (and Cd(II) and Pb(II) [40]. It was found that maximum adsorption occurred at pH 7, and the spontaneous chemisorption data fitted well with Langmuir model. Mercury(II) uptake was 5.85 mmol/g (11.70, and 6.12 mmol/g for Cd(II), and Pb(II). Desorption was performed at pH 2, with a continuous decrease of Hg(II) adsorption from 5.85 mmol/g (1st cycle) to 3.19 mmol/g (fifth cycle).

Two nanoadsorbents of Hg(II) were synthesized [41]. They were of magnetic graphene oxide (MGO–NH-SH) and magnetic nanoparticle (Fe$_3$O$_4$@SiO–NH-SH), and both functionalized with a thiol group, acetylcysteine. Their capacities to Hg(II) uptake fitted well with the Langmuir isotherm model (maximum capacity 1.04 and 0.88 mmol/g for MGO and Fe$_3$O$_4$ adsorbents, respectively), following the pseudo-second-order kinetics. As eluent solutions, 0.8 M HCl (MGO based) or 2 M HNO$_3$ (Fe$_3$O$_4$ based) were used, showing both adsorbents a continuous decrease in Hg(II) adsorption after ten cycles of continuous use, being this decrease more evident in the case of Fe$_3$O$_4$-based adsorbent agent, the above was attributable to the destruction of thiol groups by the use of a strong oxidizing chemical as nitric acid is. Under optimal conditions (pH 7 and pH 6 for the respective adsorbents), the results of Hg(II)

removal from real wastewater and polluted water by MGO–NH-SH exhibited a higher adsorption capacity, reuse, and regeneration, and as well lower cost and greenhouse gas emissions compared to Fe$_3$O$_4$@SiO-NH-SH. Mercury(II) was removed preferentially to Ca(II), Cd(II), Co(II), Mg(II), Ni(II), Zn(II), K(I), Na(I), however Cu(II) and Pb(II) were removed with yields of, or around, 50%.

A polyvinylbutyral-silica composite was synthesized via sol-gel route and under microwave conditions for removal of mercury(II) from aqueous solutions [42]. The mercury adsorption was studied at various adsorption parameters. The equilibrium was achieved, with near 95% mercury removal, after 24 h at pH 8 and at a temperature of 40 °C.

A lignin xanthate resin (LXR) intercalated bentonite clay composite (LXR-BT) for the adsorption of Hg(II) (and organic doxycycline hydrochloride (DCH) antibiotic) [43]. The adsorption performance of DCH/Hg(II) by LXR-BT was investigated under various experimental parameters: dosage, solution pH, contact time, and initial DCH/Hg(II) concentration. Experimental results showed that the pH (3–8) had little influence on Hg(II) adsorption, and the adsorption capacity (0.186 mmol/g) of DCH/Hg(II) on LXR-BT was much higher than that on bentonite (0.10 mmol/g), following the kinetics and isotherms data the pseudo-second-order and Langmuir models, respectively. X-ray photoelectron spectroscopy (XPS) analysis allowed to confirm that the adsorption mechanisms of Hg(II) (or DCH) was mainly due to π-π interaction and hydrogen bonding interaction of the complexation of Hg(II) (or DCH) with the functional groups (i.e., xanthate, carboxyl, hydroxyl) presented in the LXR-BT. No desorption data were included in the published manuscript.

Acrylamide and acrylonitrile were grafted on psyllium employing as initiator CAN (ceric ammonium nitrate), and under nitrogen atmosphere to yield an adsorbent of mercury(II) [44]. The adsorbent was further optimized to obtain the highest Hg(II) uptake. Mercury(II) was removed from the solution in the pH 2–6 range, though some decreasing in the adsorption can be observed at pH 6. Equilibrium data obeyed the Langmuir adsorption isotherm (maximum mercury(II) adsorption of 0.28 mmol/g), whereas the kinetic data followed the second-order model, indicative of a chemisorption mechanism.

In the following reference [45], breakthrough curves taken from the literature were analyzed for the dynamic adsorption of Hg(II) on granulated carbon and activated carbon cloth. The next dependence on time was used:

$$\ln\left(\frac{[\text{Hg}]_{\text{in}}}{[\text{Hg}]_{\text{out}}} - 1\right) \tag{12}$$

and it was concluded that the above relationship was linear for both adsorbents. In this relationship, $[\text{Hg}]_{\text{in}}$ and $[\text{Hg}]_{\text{out}}$ were the mercury concentrations in water flows at the inlet and outlet of the adsorption column, respectively.

Carboxymethylchitosan-g-polymethylmethacrylate (CMCS-g-PMMA) was synthesized for Hg(II) adsorption from aqueous solutions [46]. The equilibrium was reached within 90 min, and the maximum metal adsorption capacity (pH 6) was found to be 0.35 and 0.44 mmol/g, from the Langmuir isotherm model at 25 °C, for chitosan (CS), carboxymethylchitosan (CMCS), and CMCS-g-PMMA, respectively. Hg (II) removal from the solution followed the pseudo-second-order model, in an exothermic and spontaneous process. Hg(II) adsorption was found to be physical. A polyethylene glycol-water treatment of the Hg(II)-loaded adsorbents (CS, CMCS, and CMCS-g-PMMA) desorbed about 96% of the metal, though a decrease in desorption performance (96 to 85%) was observed from the 1st to the 3rd cycle.

Different experimental conditions were investigated to promote the oxidation and grafting reaction on the surface of a biochar [47]. It was shown that the nitric acid pre-oxidation concentration step (NAPOC) stage was a key step to enhance the formation of oxygen-containing initiators, thus, at higher NAPOC, carbonyl, carboxyl, and hydroxyl groups were formed and resulting in the formation of mesoporous biomass-derived biochar (MBB); all of them facilitated the decoration of N$_2$ active sites of amide and pyridine under treatment of MBB with ethylenediamine (EDA), facilitating the adsorption of Hg(II) in the carbon lattice of MBB. It was found, that the increase of temperature increased

mercury(II) adsorption, being this related to the Langmuir isotherm. The product MBB-25-EDA had the highest Hg(II) adsorption capacity of 153 mg/g at pH 6, thus, 25% was the optimum NAPOC for the introduction of $N_2/O_2$ functional binding sites to remove mercury(II). No desorption data were included in the published paper.

Diethylenetriaminepentaacetic acid (DTPA)-modified cellulose adsorbent was prepared using N-[3-(trimethoxysilyl)propyl]ethylenediamine] as a crosslinking reagent, the resulting material, properly characterized, were tested as Hg(II) adsorbent from aqueous solutions [48] Different operational variables were investigated, being mercury(II) adsorption increased with the increase of the aqueous pH value in the 2–5 range, and being this increase attributable to the presence of carboxyl and amino groups in the adsorbent; however, at pH of 6, the metal adsorption decreased. Experimental results confirmed that the adsorption responded well to the Langmuir isotherm model, with a maximum Hg(II) uptake of 2.37 mmol/g, and also to the pseudo-second-order kinetic model. An 1 M HCl solution was used to remove Hg(II) from the loaded adsorbent, but after seven cycles, the adsorption dramatically decreased from 1.37 mmol/g to 0.77 mmol/g. Modified porous cellulose beads (MCBs), sizes ranging from 2 to 3 mm, were fabricated via acidic precipitation from the cellulose dissolved in ionic liquid, and surface-grafting with aminoguanidine hydrochloride using glutaric anhydride as a coupling agent. The nano-sized calcium carbonate (30% wt on dry cellulose) was added to cellulose solution prior to the precipitation in acidic medium in an attempt to promote the formation of the pore structure of cellulose beads. Modified cellulose powders (MCPs) (size 150 μm) were prepared by precipitating the solution of the cellulose which reacted with glutaric anhydride homogeneously in ionic liquid, followed by grafting with aminoguanidine hydrochloride. They were used to adsorb Hg(II) (and Cu(II)) [49]. Maximum Hg(II) uptake was reached at a pH value of 5, and the pseudo-second-order kinetic model and the Langmuir isotherm conveniently described metal uptake onto the adsorbents, which in the case of MCBs presented a maximum adsorption capacity of 2.90 mmol/g (1.49 mmol/g in the case of copper(II)); whereas the maximum adsorption capacity of MCPs was 3.12 mmol/g (1.56mmol/g for copper(II)). Using both adsorbents, Hg(II) uptake increased with the increase of the temperature. A 1 M nitric acid solution was used as eluent, and up to ten cycles, there was a continuous decrease on adsorption and desorption capacities.

A thiol-functionalized metal-organic framework (SH-MiL-68(In)) was prepared through a post-synthesis modification procedure, and was used as Hg(II) adsorbent from water [50]. Maximum adsorption was at pH 4 with a Hg(II) adsorption capacity of 2.24 mmol/g. The adsorptive performance resulted from the strong binding interactions between -SH soft basic groups and Hg(II) soft acidic ions, and responded to the next reaction:

$$-SH + Hg^{2+} \rightarrow -S - Hg^+ + H^+ \tag{13}$$

In the presence of others accompanying metals, Hg(II) was quantitatively adsorbed, whereas Cd(II), Co(II), Mg(II), Cu(II), Mn(II), Ni(II), Pb(II), and Zn(II) were adsorbed with a yield of less than 10% each. The adsorbent was regenerated through washing it with a $10^{-2}$ M HCl, 10% thiourea, and deionized water. It was described that, after 5 cycles, there was no loss in the Hg(II) adsorption capacity. A biosorbent, cysteamine-modified cellulose nanocrystals (Cys-CNCs), was synthesized by a mild periodate oxidation of cellulose nanocrystals, and further grafting with cysteamine and used for adsorption of Hg(II) from aqueous solutions [51]. As in the previous reference, the Hard–Soft-Acid–Base theory was used to improve the adsorption of this toxic element. Under various experimental conditions, whereas maximum adsorption was reached at pH 5, it was shown that the pseudo-second order model described the adsorption kinetics, and also the Langmuir model of monolayer adsorption, the maximum adsorption capacity was 4.23 mmol/g. The adsorbent was highly selective for Hg(II) adsorption in the presence of coexisting ions (Table 3).

**Table 3.** Percentage of metal removal using various cysteamine-modified cellulose nanocrystal (Cys-CNC) adsorbents.

| Element | Cys-CNC$_{1.84}$ | Cys-CNC$_{2.25}$ | Cys-CNC$_{4.05}$ |
|---|---|---|---|
| Hg(II) | 75 | 89 | 98 |
| Cd(II) | 6.5 | 7.4 | 20 |
| Pb(II) | 2.9 | 6.1 | 8.4 |
| Zn(II) | 4.3 | 3.6 | 6.7 |
| Cu(II) | 3.0 | 4.3 | 15 |

Aqueous solution: Hg(II) (110 mg/L), Cd(II) (108 mg/L), Pb(II) (103 mg/L), Zn(II) (114 mg/L), Cu(II) (125 mg/L) at pH 5. Temperature: 25 °C. Time: 12 h.

In this system, the eluent solution was formed by a mixture of 2 M HCl and 0.5 M thiourea, and there was a slight decrease in Hg(II) adsorption from the 1st cycle (quantitative adsorption) to the 4th cycle (near 90% adsorption).

A biosorbent, l-cysteine modified cellulose nanocrystals (Lcys-CNCs), was synthesized by functionalizing high surface area cellulose nanocrystals with l-cysteine using periodate oxidation and reductive amination reaction, and the product was used for Hg(II) removal through strong mercury chelating groups [52]. Again, the pseudo-second order model and the Langmuir model of monolayer adsorption described the metal uptake, with a maximum concentration of 923 mg/g. The adsorbent material removed Hg(II) in the presence of Cd(II), Pb(II), Cu(II) and Zn(II), with a yield of 87%, which was lower that the 93% obtained from mono-elemental Hg(II) solutions. Moreover, and after four cycles of adsorption-desorption the percentage of Hg(II) decreased from 93% until 55%.

The oxidative polymerization of 1.5-diaminonaphthalene (DAN) with ammonium persulfate yields poly(1,5-diaminonaphthalene) microparticles (APS) with abundant reactive amino and imino groups on their surface [53]. The existence of the microparticles gave to this material electrical semiconductivity, high resistance to strong acid and alkali media, and strong adsorption properties towards Hg(II) (also Pb(II) and Ag(I)). Whereas 1.5DAN adsorbed in the order Hg(II) (94%) > Ag(I) (84%) > Pb(II) (0%), the increase of APS changed the order to Hg(II) > Pb(II) > Ag(I). The adsorption of Hg(II) was related to a chelation, redox and ion exchange process. Elution was performed with 0.5 M nitric acid, and also the adsorbent was regenerated, though no data were included in the published manuscript.

An azine-linked covalent organic framework (ACOF) was synthesized and proposed for the adsorption of Hg(II) (and U(VI)) [54], with maximum Hg(II) uptake of 0.87 mmol/g (0.71 mmol/g for U(VI)). The adsorption increased through pH 2 to pH 12, which was related to the presence of soluble Hg(II) species, i.e., Hg(OH)$_2$. Tautomerization between the enol-form and keto-form was verified in the structure of the ACOF which can be induced by the adsorption of metal ions and also affect the adsorption properties simultaneously. No data were included on desorption step. A strategy for the design and synthesis of covalent organic frameworks (COF) with flexible alkyl-amine as a building block and intramolecular hydrogen bonding as a knot in the network was proposed [55]. Thus, by using 1,3,5-triformylphloroglucinol and oxalyldihydrazide (ODH) as initiators, a COF material, TpODH, in which various organic building units were combined, through hydrazone bonds, to form two-dimensional porous frameworks was developed. The as-synthesized materials presented a Hg(II) adsorption capacity of 8.43 mmol/g (1.61 mmol/g for Cu(II)), values which corresponded to adsorption efficiencies exceeding 99%; Pb(II), Cd(II), and Cr(II) were adsorbed with less than 5% efficiency. The adsorption of Hg(II) relied on synergistic combination of electrostatic and coordination interactions. No data about desorption. Bagasse was converted into a carbon composite featuring hierarchically porous structure with a large surface area (351 m$^2$/g). The adsorbent contained iron oxide and manganese oxide (Fe$_2$O$_3$ and MnO$_2$ as dispersed phases). This composite was used to adsorb Hg(II) [56], with an adsorption capacity of 0.05 mmol/g, and best adsorption efficiencies in the 5–7 pH range. No desorption information included in the manuscript.

2,6-diaminopyridine (PD) and polyamine compounds: ethylenediamine (EDA), triethylenetetramine (TETA), and tetraethylenepentamine (TEPA) were used to modify chitosan (CS), and all the derivatives were used to adsorb Hg(II) from aqueous solutions [57]. The results confirmed that successful modification improves the Hg(II) adsorption significantly compared to pristine CS. In all the cases, Hg(II) uptake reached maxima at pH values above 4, being the metal adsorption driven by the pseudo-second-order kinetic model, whereas the Langmuir isotherm fitted the Hg(II) adsorption. Maximum adsorption capacities for Hg(II) were 0.86, 1.52, 1.37, and 1.15 mmol/g for PD-CS, PD-EDA-CS, PD-TETA-CS, and PD-TEPA-CS, respectively. Hg(II) adsorption was attributed to a combination of electrostatic attraction and a coordination reaction. Desorption was performed with 0.01 M Na$_2$EDTA, however, Hg(II) adsorption continuously decreased after repeated use, i.e., quantitative adsorption in the 1st cycle, 85–88% in the fifth cycle.

In order to investigate its suitability as Hg(II) adsorbent, a material was prepared via modifying corn husk leaves with bismuthiol I [58]. Experimental date indicated that the percentage of Hg(II) removal was near 99% at pH 1.0–7.0, and the adsorption process followed a Hill isotherm model, with a maximum metal uptake of 716 mg/g:

$$[Hg]_{a,e} = \frac{nN_M}{1 + \left(\frac{[Hg]_{1/2}}{[Hg]_{aq,e}}\right)^n} \tag{14}$$

where n was the amount of metal ion per site, N$_M$ the density of the receptor sites, [Hg]$_{1/2}$ and [Hg]$_{aq,e}$ represented the concentration of mercury(II) at half equilibrium and saturation, respectively. The equilibrium adsorption capacity was estimated as:

$$[Hg]_m = nN_M \tag{15}$$

The adsorption process proceeded by chelation between nitrogen/sulfur groups and Hg(II). In a simulated wastewater containing Hg(II), Cd(II), Co(II), Ni(II), Zn(II), Mg(II), Mn, and As, mercury removal efficiency was greater than 99%, whereas for the other accompanying metals the values were 9, 5, 17, 5, 11, 3, and 8, respectively. Elution was performed by the use of solutions containing 10% thiourea and 2% HCl, and after three cycles, Hg(II) recovery was beyond 99%. A novel trithiocyanuric acid-modified corn bract (TCA-CCB) was prepared, and its applicability on Hg(II) removal was investigated [59]. The adsorption process can be best described by the pseudo-second-order kinetic equation and Hill isotherm model. The material had an adsorption capacity of 1.94 mmol/g, being the Hg(II) adsorption onto the adsorbent related to chelation and ion exchange between amino/thiol groups and the metal. Equally that in the above reference, Hg(II) was adsorbed preferably to Cd(II), Co(II), Ni(II), and Zn(II), in this case it was mentioned that the adsorption was performed at a pH of 3. Desorption step was done using the same desorption solution that in the previous reference.

A monolithic adsorption material (ZnS-ZIF-8) was prepared by means of the functionalized filter paper and it was used to Hg (II) capture from wastewater [60]. From a solution containing 0.2 g/L Hg(II), near quantitative adsorption was reached in the 2–7 pH range. The adsorption was fitted to the Langmuir model (Equation (3)) with a maximum adsorption of 4.62 mmol/g. Hg(II) was adsorbed in preference of a number of ions, including Cu(II), Ba(II), Ni(II), Mn(II), K(I), and Zn(II). Hg(II) can be eluted from Hg(II)-loaded adsorbent with 1 M Na$_2$S solution, and apparently, after the reaction with Na$_2$S, the adsorbent was synthesized again. A biomimetic SiO2@chitosan composite, presenting a large amount of active sites such as amino and hydroxyl groups, was used as adsorbent material towards heavy metal ions, including Hg(II) (and As(V)) [61]. Following the Langmuir model, the composite had a maximum uptake of 1.02 mmol/g from Hg(II) (2.669 mmol/g in the case of As(V)). No desorption data included in the published work.

ZrO$_2$ based metal-organic frameworks (MOF-808) were synthesized by a sol-gel method, and amidoxime (AO) functional groups were subsequently grafted to the MOF-808 surface by a wet

chemistry process, resulting in MOF-808/AO [62]. Both products were used in the adsorption of Hg(II) from aqueous solution, and under different experimental variables such as, pH, temperature, reaction time, and initial metal concentration. The experimental data showed that the grafted material increased the metal uptake to 1.91 mmol/g against 1.71 mmol/g of the MOF-808 adsorbent (Figure 2).

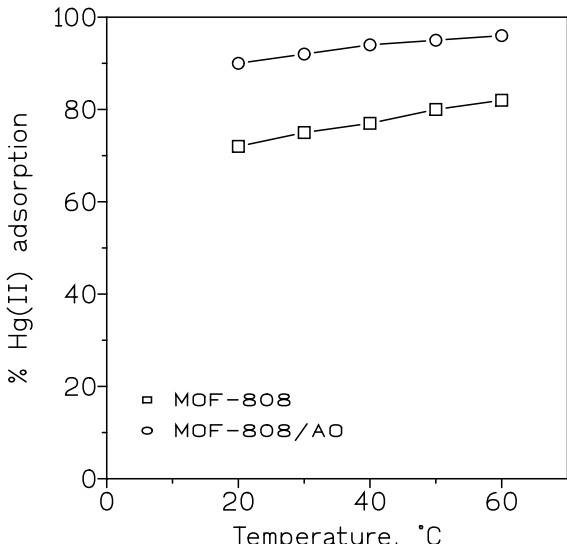

**Figure 2.** Percentage of Hg(II) adsorption at various temperatures. Aqueous phase: 0.2 g/L Hg(II) at pH 5. Adsorbents dosage: 5 g/L. Time: 80 min.

This increase was attributable to the formation of a complex with strong bonding interactions between AO groups, presented in the grafted adsorbent, and Hg(II). The adsorption process was endothermic. Desorption experiments were not included in the work.

The removal of heavy metal such as, Hg(II) (and Pb(II), Cd(II), and Cr(III)) from wastewater, using a synthetic bioadsorbent made of gelatin-chitosan (GC) hydrogel particles and operated at neutral pH [63] and silica-gelatin aerogels [64] was investigated. The hydrogel particles presented a higher affinity towards Hg(II) against that observed with the other metals investigated in the work. There was not included data about the desorption step. Supercritically dried, mesoporous silica−gelatin hybrid aerogels of 4–24 wt % gelatin content show high selectivity for the adsorption of aqueous Hg(II) in the simultaneous presence of Cu(II), Cd(II), Co(II), Pb(II), Ni(II), Ag(I), and Zn(II).

A three-dimensional (3D) graphene as reduced graphene oxide hydrogel (rGH)-encapsulated silica gel (SG-PEI/rGH) was prepared by a moderate chemical reduction strategy using ascorbic acid [64]. This composite structure was used to investigate its performance against the removal of Hg(II) from aqueous solutions. Its adsorption capacity was 266 mg/g, which represented an increase of 32% if compared with the silica gel [65]. This increment was attributable to the formation of a N–Hg complex with multi-amino groups on the surface of polyethyleneimine-modified silica gel (SG-PEI), coupled with the rapid diffusion of Hg(II) due to the rGH network structure. In this work, the eluent solution was composed of HCl (1 M), thiourea (2%), and cysteine (0.1 M), experimental results showed that there was a continuous decrease in the removal efficiency after consecutive adsorption-desorption cycles, i.e., near 100% in the first adsorption and 75% in the fifth cycle.

Pyrrhotite (MPy), derived from the thermal activation of natural pyrite, was used for the removal of Hg(II) from aqueous solutions [66]. The adsorption was related to the Langmuir model, with a maximum uptake of 0.83 mmol/g at pH 6, and fitted to the pseudo-second-order model. Moreover, the adsorption was described to be spontaneous and endothermic. Hg(II) uptake was mainly attributed to a chemical reaction resulting in cinnabar formation and the electrostatic attraction between the negative charges in MPy and positive charges of Hg(II).

A carboxy-functionalized covalent organic framework, COOH@COF, was synthesized and used in the adsorption of $Hg^{2+}$ (and $Pb^{2+}$) from waters [67]. The adsorption efficiency increased with the pH (2 to 6), and with the temperature (10–25 °C), however in the 25–40 °C range, Hg(II) uptake remained almost constant. This COOH@COF had adsorption capacities of 0.49 mmol/g for Hg(II) and 0.60 mmol/g in the case of Pb(II). The adsorption isotherm followed the Langmuir model, and the kinetics fitted bothe the pseudo-first and the pseudo-second-order models. Desorption was performed by the use of 6 M HCl solutions. The adsorbent can be recycled 20 times with a slight but continuous loss of the adsorption capacity (from 0.50 mmol/g to 0.45 mmol/g), but this capacity completely fell down after 25 cycles (0.25 mmol/g).

The adsorption property of magnetic graphene oxide grafted 3.0 generations polymaleicamide dendrimer ($GO/Fe_3O_4$-g-G3.0) nanohybrids towards Hg(II) in aqueous solution was investigated [68]. Magnetic separation technology was also used in the process. The results showed that Hg(II) was partially reduced to Hg(I) in the adsorption process, whereas the pH value had a major impact on Hg(II) adsorption, increasing the metal adsorption with the increase of the temperature (5–45 °C). The process was well fitted to the pseudo-second-order kinetic and the Langmuir adsorption isotherm models. No data about the desorption step were included in the work.

Michael addition and amidation condensation reaction with, (i) carrier: hexamethylenediamine (HMD)-modified magnetic graphene oxide ($HMGO-NH_2$), and (ii) function monomers: HMD and methyl acrylate, was used to generate MGO grafted with longer hydrophobic chain length low generation HMD type polyamidoamine dendrimers (HMGO-PAMAM-G1.0) [69], and the material was characterized by the usual technologies. Further, it was used to adsorb Hg(II) (and Pb(II)) from aqueous solutions. Metals uptake were in agreement with the pseudo-second-order kinetic model and the Langmuir isotherm (Equation (3)); however, Hg(II) was partially reduced to Hg(I) in the adsorption process. The desorption step was not considered in this work.

Iron oxide fine waste by-product from steel industry was converted into nanoparticulates ($Fe_2O_3$ NPs) and further crosslinked with starch, as a good stabilizer and biodegradable polymer, to transform the above material to $Fe_2O_3$ NPs-starch nanocomposite [70]. The adsorption behavior of this nanocomposite was investigated against the presence of Hg(II) (also Pb(II) and Cd(II)) in aqueous solutions, and under various experimental conditions: pH, contact time, nanocomposite dosage, and metal concentration Mercury(II) adsorption increased from pH 1 to 7 (Figure 3), being the maximum Hg(II) uptake of 0.66 mmol/g, (9.65 mmol/g for Pb(II), and 2.87 mmol/g for Cd(II)). The results showed that Hg(II) adsorption best fitted to all models. The published paper did not give any information about the desorption step.

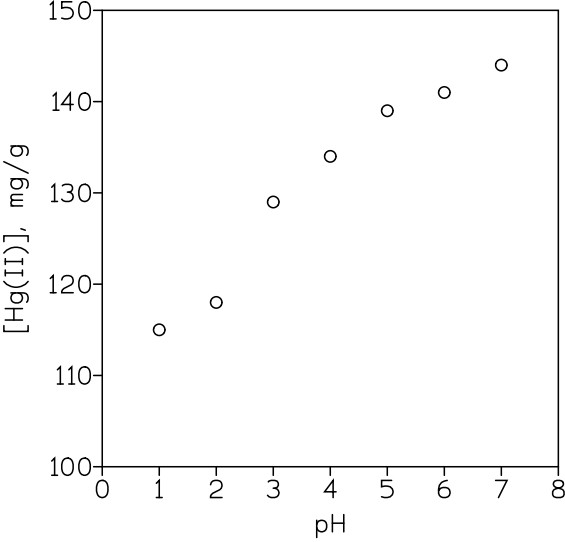

**Figure 3.** Effect of pH on Hg(II) adsorption.

A method for encapsulation and functionalization of nanozero valent iron particles (NZVI) with 3-aminopropyltrimethoxysilane ($NH_2$) and 2-pyridinecarboxaldehyde (PY), to produce the nanocomposite (NZVI-$NH_2$-PY) was developed [71]. In addition, NZVI nanoparticles were functionalized with ethylenediamine (ED) and PY to produce NZVI-ED-PY nanocomposite. These two magnetic nanocomposites had been used for removal of Hg(II) and other heavy metals (Co(II), Zn(II), Pb(II), Cd(II), and Cu(II)), besides radioactive isotopes ($^{65}$Zn and $^{60}$Co) from water. Experimental results showed that the increase of the pH from 1 to 7 increased mercury(II) adsorption, and NZVI-$NH_2$-PY nanocomposite was more selective toward Hg(II) (and Pb(II) and Cd(II)), while the selectivity of NZVI-ED-PY with respect to Hg(II) was not as good as that showed by the other nanocomposite. Different kinetic models were applied and the investigated metal ions were characterized to undergo the pseudo-second order using both NZVI-NH2-PY and NZVI-ED-PY nanocomposites. There were not included data about Hg(II) desorption from the loaded adsorbent.

L-Cystine functionalized δ-FeOOH nanoparticles (Cys-δ-FeOOH) were synthesized and used as an adsorbent of Hg(II) in aqueous solution [72]. The compound had a point of zero charge of 5.7 which seemed to be adequate for the adsorption of Hg(II) near neutral pH values. The maximum Hg(II) uptake of Cys-δ-FeOOH at pH 7 were found to be 1.08 mmol/g, which compared very well with that of 0.17 mmol/g of δ-FeOOH. Kinetics data were best fitted to the pseudo-second-order model, suggesting chemical adsorption on the surface and pores of the nanoparticles. Potassium iodide was used as eluent for mercury. δ-FeOOH and Cys-δ-FeOOH filters were constructed for purifying mercury-contaminated water. The filters were highly efficient to treat mercury-contaminated water from Doce river (Brazil), with a content of 30 μg/L of the metal, using the cys-δ-FeOOH filter, the exiting water contained less than 1 μg/L, which apparently reducing the concentration of mercury in water to values below the allowed limits by the current legislation.

Bentonite clay was employed to synthesize geopolymer that can remove Hg(II) and other heavy metals (Cu(II), Pb(II), Ni(II), and Cd(II)) from industrial wastewaters. $Fe_3O_4$ nanoparticles were used to modify the geopolymer and the use of the geopolymer/$Fe_3O_4$ nanocomposite as a magnetic adsorbent for heavy metals removal from aqueous solution was investigated [73]. No desorption data were included in the manuscript. The prepared magnetic geopolymer showed a 92% removal efficiency in the case of Hg(II) from the industrial wastewater.

Chitosan modified polyurethane foam (Chi-PUF) was investigated to remove Hg(II) from synthetic solutions as well as Hg(II)-contaminated well water samples from Aceh Jaya Regency Indonesia) [74]. Preliminary experiments fixed the equilibration time and pH as 60 min and 7, respectively, with data fitted to the Freundlich isotherm model. The application of the adsorbent on the well water samples indicated that about 80% Hg(II) removal efficiency was achieved. No data about Hg(II) desorption were included in the work.

Chemical modified lignin is prepared with diethylenetriamine and then used to prepare lignin derivate magnetic hydrogel microspheres (LDMHMs) via blending with $Fe_3O_4$. The adsorbent was used to remove organic dyes and heavy metals from aqueous solutions [75]. In the case of Hg(II), best adsorption values were obtained at pH of 5, with maximum uptake was 0.27 mmol/g, value which was evidently higher than the obtained, 0.02 mmol/g, in the case of unmodified lignin. Desorption step was carried out using 5 mL of deionized water (pH 7) and vibration. The adsorbents were recycled by magnetic separation, regenerating from acid condition and reused for multiple cycles, though from the third cycle, a decrease in the adsorption capacity was observed.

Activated carbon from *Bambusa vulgaris* var. *striata* was produced by chemical activation with NaOH, and used for mercury(II) adsorption [76]. The mechanism of the adsorption process through a fixed-bed column was fitted to the Thomas model:

$$\frac{[Hg]_{aq,t}}{[Hg]_{aq,0}} = \frac{1}{1 + \exp\left(\left(k_T[Hg]_a \frac{m}{F}\right) - \left(k_T[Hg]_{aq,0} t\right)\right)} \tag{16}$$

where $[Hg]_{aq,t}$ and $[Hg]_{aq,0}$ were the mercury concentrations in the aqueous solution at time t and time zero, respectively, m was the adsorbent mass in the column, F was the feed volumetric flow rate, $[Hg]_0$ was the adsorption capacity of the adsorbent, $k_T$ was the rate constant, and t the elapsed time. Experimental data showed that the adsorption capacity was of 1.09 mmol/g. Again, desorption data were not given in the manuscript

Guar gum (GG)-g-(acrylic acid (AA)-co-3-acrylamido propanoic acid (AMPA)-co-acrylamide (AM))-g-cow buffing dust (CBD)/(GGTPCBD), a smart carbohydrate and protein-based doubly-grafted interpenetrating terpolymer hydrogel, was synthesized and used in the adsorption, among others inorganic and organic pollutants, of Hg(II) [77]. The adsorption was related to the pseudo-second order model and the Langmuir isotherm in a thermodynamically spontaneous chemisorption process. The maximum uptake was 859 mg/g at 30 °C. Desorption was carried out at pH 7. After continuous use (5 cycles), Hg(II) adsorption capacity decreased from 0.72 to 80.41 mmol/g.

By consideration of the usability of hypercrosslinked polymers, porous materials presenting chelating thiophene units in the pore walls were synthesize [78]. Including in these porous materials, Th-2 had high efficiency in the removal of Hg(II) aqueous solution, with a metal uptake of 0.72 mmol/g, and rendering mercury(II) concentrations in the drinking water below the acceptable value. The material adsorbed the heavy metal because of the synergistic action between densely packed thiophene sites and rapid diffusion across the micropores and interparticle mesopores. Then, 10% thiourea in 0.05 M HCl was used as eluent solution, and the adsorbent maintained its adsorption capacity after six cycles.

Graphene oxide nanoparticles were thiol-functionalized, magnetized and used to remove Hg(II) [79]. The maximum adsorbing capacity of thiol-functionalized graphene oxide was estimated to be 0.65 mmol/g by Langmuir model in a spontaneous process. Using solutions containing 10–50 mg/L Hg(II), the kinetics fitted to the pseudo-second order model. As eluent, HCl solutions gave better results than nitric or sulphuric ones. The Hg(II)-loaded nanoparticles were separated from the liquid phase by magnetic field, and after three cycles, using 0.5 M HCl solution as eluent, there was a decrease in the adsorption efficiency from 96% to 75%.

An interpenetrating tetrapolymer-hydrogel was used to investigate its adsorption properties on Hg(II) (also Bi(III) and brilliant green-crystal violet) [80]. The chemisorption data for Hg(II) best fitted with Langmuir and combined Langmuir-Freundlich isotherm models for monoelemental and binary adsorption, respectively. In the case of the monoelemental solution, and at 20 °C, Hg(II) uptake was 4.78 mmol/g, being this valor decreased until 3.78 mmol/g in the case of the binary solution. Adsorption and desorption steps were carried out at pH values of 7 and 2, respectively, and after five cycles, the continuous decrease (5.42, 4.93, 4.54, 4.22, and 3.97 mmol/g) in Hg(II) adsorption capacity demonstrated the continuous degeneration of the adsorbent.

A thioether-functionalized covalent triazine nanosphere, SCTN-1, had been used for the removal of Hg(II) (and elemental mercury) from contaminated water [81]. Its adsorption capacity was of 6.25 mmol/g for Hg(II) (4.05 mmol/g for elemental mercury), and while it was maintained at pH of 6 and above, decreased at low pH values. The system followed the pseudo-second-order kinetic model. Solutions of HCl (6 M) were used to elute the adsorbed Hg(II); after seven cycles, the system maintained its adsorption characteristics, though a small decreased began to be observed.

Using emulsion templating technique, hierarchically porous poly(vinylsulfonic acid) beads were produced (PVSA), which were then used for the in situ production of silver nanoparticles yielding poly(vinylsulfonic acid)-Ag nanocomposite (PVSA-Ag NC) beads. Due to its characteristics, the nanocomposite beads (1.56 ± 0.20–1.50 ± 0.14 mm) were used to remove Hg(II) and RhB and to kill *Escherichia coli* (Gram-negative) and *Staphylococcus aureus* (Gram-positive) bacteria [82]. The adsorption capacities of these materials for Hg(II) were in the 0.84–0.951 mmol/g range. After three cycles, the adsorption capacity decrease about 40% (PVSA-Ag NC) or 22% (PVSA) of their initial capacity.

Magnetic bio-composite (CMNC) was fabricated using curcumin and $Fe_3O_4$ nanoparticles [83]. Mercury(II) uptake fitted to the Langmuir adsorption isotherm, with a maximum capacity of 145 mg/g,

being this capacity compared with that of other adsorbents (Table 4). The CMNC adsorption capacity was related to the various amino and oxygen-containing groups on the adsorbent surface which bond $Hg^{2+}$ with relative easiness.

**Table 4.** Comparison of monolayer adsorption capacity of various adsorbents.

| Adsorbent | $[Hg(II)]_{max}$ (mmol/g) |
|---|---|
| CMNC | 0.72 |
| Pistachio-nut/licorice-residues | 0.73 |
| Magnetic mesoporous silica composites | 0.10 |
| Bamboo | 0.01 |

The adsorption kinetic was defined by the pseudo-first-order model. With a maximum adsorption at pH 6, this capacity decreased from 96% to 65% when the Hg(II) concentration in the solution increased from 0.01 to 0.06 g/L. Moreover, the adsorption process was spontaneous and exothermic. Hg(II) loaded onto the adsorbent can be desorbed using 0.1 M hydrochloric acid solution at 25 °C. After four cycles the adsorption yield decreased from 85% to 75% (adsorption) and 99% to 80% (desorption).

Nanoparticles (NPs) based on thiol-functionalized chitosan (CS) was generated using capillary microfluidic (MF) device combined with ionic gelation, and were used to remove Hg(II) from aqueous solutions [84]. In addition, CS was functionalized with epichlorohydrin/cysteaminium chloride (2.73 M ratio) and further fabrication of NPs via MF and bulk mixing (BM) methods. Isotherm experiments were done with Hg(II) solutions containing 40 to 170 mg/L of the element at pH 6, and adsorbent dosage of 100 mg/L. The fit to the Freundlich isotherm indicated that multi-layer adsorption occurred on the heterogeneous surface. Hg(II) can be desorbed using 0.5 M HCl solution (Abstract section). After four cycles, the adsorption decreased from near 3.99 mmol/g to 3.24 mmol/g, whereas desorption reduced from near quantitative to 80%. An ultrasound-assisted co-precipitation method was developed to prepare lipophilic thiol-functionalized $Fe_3O_4$ magnetic nanoparticles ($EDT$-$Fe_3O_4MNPs$), with sodium dodecyl sulfonate (SDS) as dispersant and 1,2-ethanedithiol (EDT) as functionalization reagent [85]. The effects of the dosage of adsorbent, initial pH of the solution and contact time on Hg(II) removal from aqueous solutions were investigated. The experimental results fitted with the Freundlich isotherm and the pseudo-first-order kinetic model. Furthermore, the adsorbent could be effectively separated using an external magnetic field and regenerated.

Chitosan/gelatin (CG) spherical hydrogel particles for the effective removal of multiple heavy metal ions were prepared [86]. The CG hydrogel particles synthesized by inverse emulsion from the aqueous solutions of chitosan, gelatin, and glutaraldehyde, and showed a maximum removal efficiency of 98% for Hg(II) from monoelemental solutions, and 90% in multiple solutions containing Hg(II), Pb(II), Cd(II), and Cr(III). No desorption data were included in the work.

Microwave alkali-modified fly ash was synthesized as an adsorbent for Hg(II) from aqueous solution [87]. The adsorption isotherm fitted to the Langmuir with a maximum uptake of 13.30 mmol/g, attributable to a monolayer adsorption; for Hg(II) initial concentrations in the aqueous solutions of 5 and 10 mg/L, the pseudo-second-order kinetic model described the entire process of adsorption, however, the pseudo-first-order kinetic equation fitted well with the initial adsorption. Desorption step was not considered in this manuscript.

A series of (G1.0, G2.0, G3.0, and G4.0) PAMAM dendrimers modified attapulgite (ATP) sorbents (G1.0–G4.0 PAMAM-ATP) were developed in order to investigate their capacities as Hg(II) adsorbents from aqueous solution [88]. Batch experiments showed that in the adsorption process near 90% Hg(II) was removed from the solution, at pH of 5, after 80 min. Metal uptake of PAMAM-ATP adsorbents followed the order of G2.0 > G1.0 > G3.0 > 4.0, while the maximum adsorption uptake increased from 0.02 mmol/g (raw ATP), 0.45 mmol/g (amine modified ATP, M-ATP), 0.96 mmol/g (polyacrylamine modified ATP, PAM-ATP) by previous similar studies to 1 mmol/g (G2.0-PAMAM-ATP). Pseudo-second-order and controlled by chemical adsorption over a whole sorption range, with Hg(II)

adsorption increased with the increase of the temperature from 10 to 30 °C, thus, the process was spontaneous and endothermic. The Langmuir isotherm fitted to the experimental data, indicating a monolayer behavior. Using 0.1 M $HNO_3$ as eluent, the maximum adsorption capacity of the regenerated G2.0-PAMAM-ATP sorbents could still be higher than 90% after five cycles, though a slight decrease in capacity along the cycles was observed.

A Goat buffing dust (GBD), an abundantly available collagenic-waste and crosslinked styrene butadiene rubber (SBR)-based scalable biocomposite was synthesized, and used to adsorb Hg(II) [89]. Adsorption kinetics data responded to the pseudo-second order model, and also Langmuir isotherm. Spontaneous chemisorption showed the maximum uptake of 1.13 mmol/g at 30 °C. After continuous cycles of adsorption (pH 7) and desorption (pH 2), the adsorption capacity decreased from 0.92 mmol/g (1st cycle) to 0.37 mmol/g (5th cycle).

Adsorption of inorganic mercury [Hg(II)] (and organic methylmercury [MeHg(II)]) onto dye-affinity agrowaste (dye-AW) was investigated [90].  The dye-affinity adsorbents were prepared by the chemical-thermal reaction between the agrowaste (AW) and dye solutions (i.e., reactive red 120 (RR), reactive black B (RB), methylene blue (MB), and methyl orange (MO)) under alkaline medium. The maximum adsorption capacity for Hg(II) onto the reactive red 120-modified AW (RR-AW) was 2.59 mmol/g for Hg(II), with maximum adsorption in the 2–8 pH range. Kinetic data followed the pseudo-second order kinetic model with the diffusion steps controlled by the film diffusion. As eluents, both 0.1 M HCl or IK were investigated, it was found that after five cycles, the adsorption capacity was reduced a 5% or 11% of the initial one using HCl or IK, respectively. The results were also obtained by using oilfield produced water (OPW) and natural gas condensate (NGC) samples. In the case of OPW, a selective adsorption sequence Zn > Hg > As > Mg > V > Ni > Ba > Cu was found. Comment from the authors of the review: there was not indication of the arsenic and vanadium oxidation state in the treated solution.

Low-cost reactive agrowaste adsorbents, CP-Pure, CP-MPTES, and CP-RR were used in a fixed-bed adsorber to investigate their performance in the removal of Hg(II) (and methylmercury) [91]. It was demonstrated experimentally that the breakthrough and saturation times were delayed with decreasing flow rate and initial concentration, and increasing bed height. Equilibrium and kinetic analyses of adsorption data indicated that it was followed the Temkin isotherm and the pseudo-second-order kinetic model, respectively. The breakthrough curve was simulated well by the Thomas and Yoon-Nelson models. Using HCl as eluent, the regeneration studies showed that the regenerated CP-Pure and CP-MPTES have a high regeneration efficiency up to third adsorption cycle, however, after five cycles, the adsorption capacity decreased from 100% to 27% for CP-Pure adsorbent, and from 100% to 8% (4th cycle) for CP-RR adsorbent.

Various formulations of a novel gold–graphene oxide–iron oxide (Au–GO–Fe3O4) hybrid nanoparticle system were created and used to adsorb Hg(II) [92]. It was reported that the maximum metal uptake was of 0.03 mmol/g and no elution experiments were reported in the manuscript.

A low-cost mercapto-modified coal gangue (CG-SH) was fabricated by modification of coal gangue (CG) with (3-mercaptopropyl) trimethoxysilane, and used in ther removal of Hg(II) (also Pb(II) and. Cd(II)) [93]. At pH values of 2–6 the capacity was of 0.45 mmol/g. Equilibrium data fitted with Langmuir isotherm, being the maximum capacity of CG-SH of 0.89 mmol/g for mercury(II), and being the adsorption described by the pseudo-second-order kinetic model in an endothermic process. There were not elution data included in the work.

Cobalt impregnated silica nanoparticles (Co–$SiO_2$) were synthesized via one pot method involving supercritical drying followed by in-situ reduction, and evaluated for the adsorption of Hg(II) (also methylene blue (MB) and trinitrotoluene (TNT)) [94]. With a maximum adsorption in the 4–6 pH range, the adsorption of Hg(II) was fitted by Langmuir isotherm, thus the metal uptake was defined by a homogenous monolayer adsorption with maximum Hg(II) uptake of 0.20 mmol/g in aphysisorption adsorption process, which was spontaneous and endothermic with increased randomness at the solid/solution interface. The magnetic Co/$SiO_2$ nanoparticles were separated after adsorption process with the help of a magnet and showed reusability up to three cycles without much loss in adsorption

efficiency. Beads of cucumber peel were used to remove heavy metals from drinking water [95]. Deionized laboratory water was spiked with seven toxic ions: arsenic, cadmium, chromium, copper, mercury, lead, and nickel at concentrations of 0.1 mg/L and kinetic studies were performed over 72 h. Results indicated that different ions contained in a multi-ion solution were biosorbed by different mechanisms and at different rates. Equilibrium biosorption for Hg(II) was 91% (Cd (90%) and Ni (67%)) at 24 h with diffusion through the pores of the bead, and being their biosorption increased with an increase in temperature. The removal of mercury(II) was pH-dependent. Cucumber peel beads removed all spiked ions from real drinking water collected near the Macraes gold mine in New Zealand. No elution data were included in the work.

Thiol (SH) and dithiocarbamate (DTC) ligands were grafted on multi-walled carbon nanotubes (MWCNTs) and were characterized by the usual procedures [96]. Hg(II) maximum adsorption capacity of MWCNT-DTC was found to be 3.5 times higher than that of MWCNT-SH at the optimum pH of 6, with the adsorption process following the pseudo-first-order and Langmuir isotherm models. Apparently, both $Hg^{2+}$ and $HgCl^+$ species were adsorbed onto the adsorbent material. Density functional theoretical calculation carried out for the interaction of Hg(II) with the same but single walled nanotubes, SWCNT-DTC and SWCNT-SH, corroborated the experimental output that former have greater adsorption than the later. In equimolar mixtures of Hg(II) and Ca, Mg, Pb, Li, Na, and Cd, the selectivity toward Hg(II) was greater in MWCNT-DTC than in MWCNT-SH. Best desorption percentages 65% (MWCNT-SH) and 95% (MWCNT-DTC) were yielded with 5% thiourea, however, the mixture of 5% thiourea and 0.01 M $HNO_3$ improved the percentage of desorption to near 100% in the case of MWCNT-SH adsorbent, but not in the case of the dithiocarbamate one.

Cow buffing dust (CBD), one of the abundantly available semisynthetic collagenic solid wastes, had been used as a nonsulfur cross-linker of natural rubber (NR) for the generation of a NRCBD-biocomposite superadsorbent [97]. This was used as adsorbent of Hg(II) and dyes (2,8-dimethyl-3,7-diamino-phenazine (i.e., safranine, SF) and (7-amino-8-phenoxazin-3-ylidene)-diethylazanium dichlorozinc dichloride (i.e., brilliant cresyl blue (BCB)). Chemisorption, via ionic interaction between NRCBD and Hg(II), was responsible for the removal of the heavy metal, whereas the process followed the pseudo-second-order kinetic model (Figure 4) and Langmuir isotherm. The spontaneous chemisorption had a maximum Hg(II) uptake of 0.83 mmol/g. Desorption step was investigated using an aqueous solution of pH 2, and the results obtained after five adsorption-desorption cycles showed a continuous degeneration of the adsorbent (NRCBD), since the percentage of Hg(II) adsorption decreased from 95% in the first cycle to 35% in the fifth cycle.

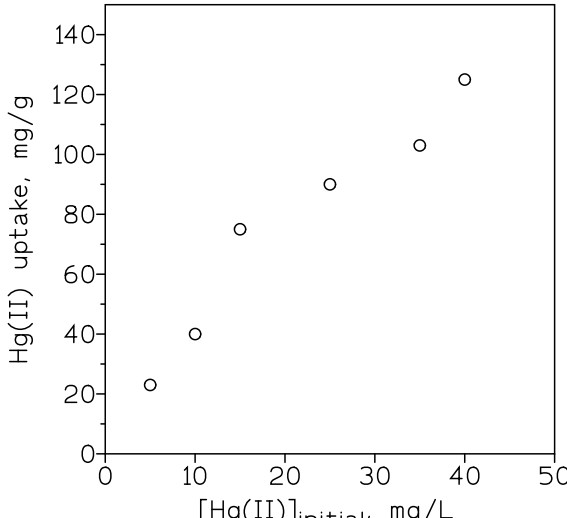

**Figure 4.** Variation of Hg(II) uptake at various initial Hg(II) concentrations in the aqueous solution. Time: 13.5 h. Results derived from the fit of experimental results to the pseudo-second order kinetic model.

The carboxy-terminated $Fe_3O_4$@polyamidoamine nanocomposites ($Fe_3O_4$@PDA@PAMAM-COOH) were synthesized, and investigated for the removal of Hg(II) (and Cu(II), Pb(II), Cd(II)) [98]. The adsorbent presented a Hg(II) maximum adsorption of 1.14 mmol/g at pH 8, following the adsorption process the pseudo-second-order kinetics and the Langmuir isotherm.

A silver nanoparticles-doped synthetic sodalite composite was synthesized, characterized, and used in the removal of Hg(II) from aqueous solutions of pH 2 [99]. The results indicated that the sodalite nanocomposites removed up to near 99% of Hg(II), which was 16% and 70% higher than the removal achieved by sodalite and parent coal fly ash, respectively. The adsorption mechanism involved: adsorption, precipitation, and $Hg^0$-$Ag^0$ amalgamation. The study of the anions effect ($Cl^-$, $NO_3^-$, $C_2H_3O_{2-}$, and $SO_4^{2-}$) indicated that the $Hg^{2+}$ uptake was comparatively higher when chloride was present in the solution. Elution data were not included in the publication.

An adsorbent derived from cocoa (*Theobroma cacao* L.) husk residual biomass chemically modified with sodium hydroxide was to Hg(II) (and Ni(II)) [100]. Particle size significantly affected the adsorption process. With an optimum particle size of 0.36 mm in the case of Hg(II) (92% adsorption). The adsorption capacity of cocoa husk residual biomass was related to the Elovich model and the Freundlich isotherm; this adsorption capacity can be improved by modification with sodium hydroxide the residual biomass.

A superparamagnetic molybdenum disulfide ($Fe_3O_4$@$MoS_2$) with an expanded interlayer spacing (1.0 nm) and high magnetic susceptibility was synthesized and investigated to adsorb Hg(II) [101]. Upon treatment with the adsorbent, over 99% of the initial Hg(II) concentration (0.015 mg/L) in the aqueous solution can be removed in 2 min. The adsorbent had also high selectivity for Hg(II) (99% adsorption) versus other eight representative metal ions: Na, K, Mn, Ni, and Cd were not adsorbed, whereas Cr, Cu, Pb were adsorbed at 14%, 24%, and 27%, respectively. Furthermore, the selectivity for heavy metal ions can be optionally tailored by simply controlling the contact time during the uptake. The outstanding Hg(II) adsorption was possible due to Hg-S complexation and ion exchange between $Hg^{2+}$ and incorporated iron(III) ions. Not elution data included in the work.

Calcium oxide (CaO) nanoparticles were prepared by the sol-gel strategy from $CaCl_2$ and calcinated the gathered powder at various temperatures (400, 500, and 600 °C), the resulting products were evaluated in the removal of Hg(II) (and Cr(II)) from solutions [102]. Every one of the analyses was completed by batch method. The impact of pH (2, 5, 7, 8, and 10), stirring time, temperature, and the presence of different foreign ions has been researched (phosphate, sulphate, chloride, magnesium, calcium, and manganese(II)). In the presence of these ions, mercury adsorption exceeding 98% from a solution of 0.01 mg/L Hg(II) at pH 7. *Lysinibacillus sphaericus* strains were used to adsorb Hg(II) from aqueous environment [103], with a maximum adsorption efficiency of 0.15 mmol/g during the first 5 min (removal over 95% of initial Hg (II). This process was escalated in a semi-batch bubbling fluidized bed reactor (BFB), using rice husk as the immobilization matrix, leading to a similar level of efficiency.

Sulfur containing ion imprinted polymers (S-IIPs) were used for the uptake of Hg(II) from aqueous solution [104]. Cysteamine, which was used as the ligand for Hg(II) complexation, was grafted along the epichlorohydrin crosslinked carboxylated carboxymethyl cellulose polymer chain through an amide reaction. The uptake process followed the pseudo-second-order kinetic equation, and the Langmuir isotherm (in the range of 25–55 °C), with maximum adsorption capacities around 0.40 mmol/g. Hg(II) adsorption was spontaneous and endothermic. The adsorbent was very selective towards Hg(II) in the presence of Cu(II), Zn(II), Co(II), Pb(II), and Cd(II); based in the separation factor the sequence was Pb > Zn > Cu > Cd > Co. 1 M nitric acid or 1% thiourea in 0.5 M HCl were effective to remove Hg(II) from the loaded adsorbent. The performance of S-IIPs was also evaluated against real samples, i.e., Hg(II) wastewater, ground water, and tap water.

Several technologies for the removal of Hg(II) from water was reviewed [105]. These included: adsorbents based on thiol, thio and dithiocarbamate groups, miscellaneous compounds, and adsorption mechanisms.

The next reference [106] explored trends and advancements in nanomaterial technology for Hg(II) removal. Conventional options: liquid-liquid extraction, precipitation, ion exchange resins, membrane separation, adsorption, and bioremediation were briefly commented. Nanoadsorbents included iron oxide-based nanomaterial, magnetite, other oxide nanoparticles and composites with transition metals (i.e., $MnO_2$ and $TiO_2$), transition metal-based nanoparticles and gold-based nanoparticles. The review compared the performance of different nanomaterials versus commercial adsorbents.

A silk fibroin-based bentonite composite was prepared to remove lead, cadmium, mercury, and chromium [107]. The mechanism of adsorption was apparently based on complex formation and ion exchange. Experimental data showed that the adsorption capacity increased with the increase of the aqueous pH values, however from pH 5, metal precipitation occurred. The adsorption percentage decreased with the increase of the initial metal concentration (1–30 mg/L) in the solution, and with the increase of the temperature (20–60 °C). The kinetic data fitted well to pseudo second order equation and Langmuir isotherm, which indicated of homogeneity of adsorption sites on the SF/clay composite. The monolayer adsorption capacity was 0.05 mmol/g for Hg(II)and 0.19mmol/g in the case of Cr(VI). No elution data was given in the text. A number of hybrid composite materials m-aramid/chitosan aimed to the removal of Hg(II) from water was synthesized by using various weight ratios of m-aramid to chitosan (100/0, 85/15, 65/35, 50/50, and 35/65) [108]. The metal adsorption capacity increased with the increase of chitosan content and reaching 0.09, 0.27, 0.43, 0.51, and 0.72 mmol/g for the above ratios, respectively. Experimental data showed that the process was described by the pseudo-second-order kinetic model, and the Langmuir and Freundlich models.

Porous polysilsesquioxane with $NH_2$ and SH bifunctional groups (PAMPSQ) coated poly(p-phenylenetherephthal amide) (PPTA) fibers adsorbents (PPTA-AM) was prepared and used to adsorb Hg(II) [109]. The PAMPSQ coated on the PPTA surface was in the form of nanoparticles and its morphology varied with the proportion of the reactants. The PAMPSQ exhibited loose open meso or macroporous features. Maximum Hg(II) uptake was achieved at pH 5, in a process controlled by the pseudo-second order kinetics model and the Langmuir isotherm. Mercury(II) was adsorbed preferentially to Pb(II), Cu(II), Ni(II), and Zn(II), with some co-adsorption in the case of Ag(I) (the above results derived from the use of binary solutions). Hg(II) was remove from the adsorbent by the use of a 0.5 M HCl solution containing 4% thiourea. After five cycles, the percentage of Hg(II) slightly decreased (91%) in comparison with the result obtained in the first cycle (95%).

A carboxylated $CoFe_2O_4@SiO_2$ was prepared by EDTA-functionalized method in a hydrothermal method [110]. The results showed that the material had a maximum adsorption capacity of 0.51 mmol/g for mercury(II) at pH 7, being the adsorption process due to a chemical reaction involving chelation (via EDTA) and single-layer adsorption; this adsorption process followed the pseudo-second-order kinetic and Langmuir adsorption isotherm models. Hg(II) uptake was a spontaneous and exothermic, and being $\Delta S^0$ negative, the order of the solid-liquid interface increased during the adsorption process. Elution was performed with a 0.1 M HCl solution, decreasing the adsorption capacity about 15% after five cycles of continuous use.

The next reference [111], reviewed the adsorption mechanism and the performance of different Hg(II) adsorbents. The Hg(II) removal capacities and related adsorption mechanisms of various adsorbents were summarized in Table 5. General considerations indicated that the advantages of physical adsorption were low cost and simple operation process, whereas chemical adsorption showed fast adsorption rate and high adsorption capacity. The combination of physical and chemical adsorption seemed to be of major interests due to the combined effects of the large surface area and functional complexation interaction in relationship with both operational modes.

**Table 5.** Selected adsorbents for Hg(II) and their related adsorption mechanisms.

| Adsorbent | Hg(II) Adsorption Capacity (mmol/g) | Type of Adsorption |
| --- | --- | --- |
| Modified Fe oxide | 0.003 | Chemical |
| Activated carbon | 0.004 | Physical |
| Fe-Sn-MnO$_x$ | 0.02 | Chemical |
| ZSM-5-zeolite | 0.26 | Physical |
| Porous carbon | 0.76 | Physical |
| Ag-Zn nanoparticles | 2.76 | Chemical |
| MWCNTs-SH | 0.53 | Both |
| Ti$_3$C$_2$T$_x$/Fe$_2$O$_3$ | 5.62 | Both |
| Nanocolloidal hydrogel (CNC-GQD) | 0.82 | Both |

A lignin-based nano-trap (LBNT) through functionalizing one of the most abundant biomass on Earth, lignin, had been synthesized and used in the adsorption of Hg(II) among other heavy metals (Ag(I), Cd(II), Cu(II), Zn(II), and Pb(II)) [112]. LBNT presented a removal efficiency greater than 99%.

Tourmaline/graphene oxide (T/GO ratio of 1:1, 1:5, 1:10) composite nanomaterials were fabricated through a hydrothermal method and examined for the removal of Hg(II) (and methylene blue) from wastewater [113]. At 35 °C, mercury(II) equilibrium adsorption capacity was of 1.46 mmol/g. The adsorption isotherm was found to fit the Langmuir model, and the adsorption kinetics fitted the pseudo-second order model.

Mesoporous silica nanoparticles with different morphologies (flower-like nanospheres with wrinkles, nanoparticles with concavities and sunken nanovesicles) were generated and functionalized with 3-mercaptopropyltrimethoxysilane (MPTS), and used for Hg(II) adsorption [114]. The material, with flower-like nanospheres morphology, exhibited highest surface area and pore volume among the above samples, and the corresponding S-H groups functionalized nanospheres showed the highest metal uptake of 2.39 mmol/g. The adsorption was represented by the Langmuir isotherm and the pseudo-second-order kinetics. Regeneration of the adsorbent can be achieved with an 1 M HCl solution containing 1% thiourea, however, the adsorbent SiO$_2$-SH-0.5 and after three cycles, lost some of its initial adsorption capacity.

A biopolymer/clay composite adsorbent was developed by introducing montmorillonite modified with sulfhydryl into a hyperbranched polyethylenimine functionalized carboxymethyl chitosan matrix (HPFC/MT-S), and used for the removal of Hg(II) from aqueous solutions [115]. The adsorption capacity was found to be of 9.35 mmol/g, also the selectivity of the composite towards Hg(II) was improved by the introduction of MT-S, yielding a high distribution coefficient value (D = 1 × 10$^7$). The improved adsorption performance was attributed to the increased porosities and chelation sites of the adsorbent. From a mixed solution containing Hg(II) (400 mg/L), Cu(II) (391 mg/L), Cd(II) (400 mg/L), and Pb(II) (397 mg/L), the adsorption capacity decreased in the order Hg(II) > Cu(II) = Pb(II) > Cd(II), with practically quantitative removal of Hg(II) from the solution. Desorption step used 2 M HCl solution, with a slight decrease in capacity after five cycles.

A novel polymer-based adsorbent of hyperbranched polyethylenimine functionalized carboxymethyl chitosan semi-interpenetrating network composite (HPFC) was fabricated and its performance towards Hg(II) removal was investigated [116]. Using a 1g/L dosage of the adsorbent, the Hg(II) concentration in the solution decreased from 798 to 0.02 mg/L, with adsorption maintained constant in the 2.5–5.5 pH range. The process followed the pseudo-second-order model, indicating chemical adsorption, with a maximum uptake of 7.95 mmol/g. Hg(II) can be selectively removed in the presence of other heavy metals: Cu(II), Cd(II), and Pb(II) (Figure 5).

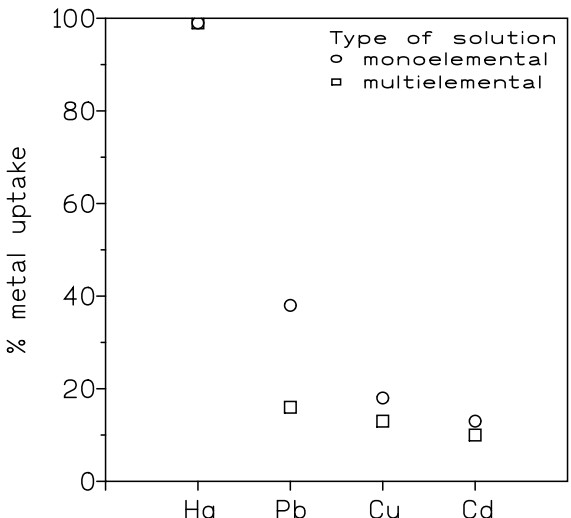

**Figure 5.** Percentages of metals uptake from monoelemental or multielemental solutions. Adsorbent dosage: 2 g/L. Temperature: 30 °C. Time: 6 h. Aqueous solutions in monoelemental solutions (pH 5.5): Hg(II): 0.80 g/L, Pb(II): 0.92 g/L, Cu(II): 0.98 g/L, or Cd(II): 1 g/L. Aqueous solution in multielemental solution (pH 5.5): Hg(II): 0.32 g/L, Pb(II): 0.37 g/L, Cu(II): 0.39 g/L, and Cd(II): 0.4 g/L.

The removal of Hg(II) by HPFC was controlled by the interaction between Hg(II) and nitrogen functional groups (i.e., amine and imine groups) of the adsorbent. Hg(II) recovery from loaded adsorbent was investigated using 2 M $HNO_3$ or 2 M KCl solutions. Using nitric acid, the adsorption efficiency decreased from near 99% (1st cycle) to almost 90% (5th cycle); however, the use of 2 M KCl solution was more critical since a mere 12% of mercury(II) was adsorbed in the 5th cycle. A chitosan/cellulose biocomposite sponge (CCS) adsorbent was developed for removing Hg(II) aqueous solution [117]. This adsorbent was generated via a glutaraldehyde cross-linking reaction and lyophilization; experiments showed that maximum adsorption occurred at pH 5.5, and the adsorption process followed the pseudo-second-order kinetic model, being the adsorption capacity of 2.47 mmol/g. The material was highly selective towards Hg(II) with respect to Cu(II), Cd(II), and Pb(II), being the coefficient of selectivity of Hg(II) 461, 35, and 227 times higher than that of the above heavy metals, respectively. 1 M nitric acid was the desorption solution, and Hg(II) uptake was 85% of its initial value after five cycles. The use of CCS was evaluated in simulated Hg(II) effluent (PVC production).

An aminopyridine functionalized magnetic $Fe_3O_4$(HO-$Fe_3O_4$@$SiO_2$-AP) adsorbent was synthesized and characterized by the usual methods to investigate its performance in the removal of Hg(II) (and Ag(I)) from solutions [118]. The adsorption process followed the pseudo-second-order kinetic model and experimental data fitted to the Langmuir isotherm, with an increment in the maximum capacity from 0.92 mmol/g (15 °C) to 222 (35 °C). In binary solutions, complete separation of Hg(II) from Cd, Cu, Ag, Zn, and Ni was achieved. Elution step used a solution of 5% thiourea in 0.2 M HCl to remove Hg(II) adsorbed onto the material, as it was being usual in these systems, there was a slight decrease of the adsorption capacity with repeated adsorption-desorption cycles. (up to three).

An adsorbent (Cys-UiO-66), derived from the reaction of $NH_2$-UiO-66 with L-cysteine, was prepared to investigate its possibilities in the removal of Hg(II) from aqueous solution [119]. The introduction of L-cysteine improved the maximum capacity from 110 mg/g of the pristine adsorbent to near 2.13 mmol/g of the derivative one. Maximum adsorption capacity was reached at pH 5, with the adsorption isotherm and kinetics models in accordance with the Langmuir and pseudo-second-order models, respectively; there was evidence that the adsorption behavior was dominated by monolayer chemisorption, and that Hg(II) formed a chelate compound with L-cysteine. The Cys-UiO-66 had better affinity for Hg(II) than other coexisting ions in wastewater and could be regenerated for at least five cycles, though the adsorption decreased a 12% of the initial one.

A magnetic nano-adsorbent ($CoFe_2O_4@SiO_2$), with core–shell structure, was functionalized with polypyrrole (Ppy) to adsorb Hg(II) from water [120]. Metal uptake obeyed the pseudo-second-order kinetic and Langmuir models, with a uptake of 3.39 mmol/g at pH 8. Experimental data indicated that the process is spontaneous and endothermic. However, Hg(II) uptake decreased as the ionic strength of the aqueous solution increased. Mercury(II) can be desorbed by 0.2 M HCl, and the adsorbent lost, after five cycles, about 13% of its initial loading capacity The adsorbent was used in the removal of Hg(II) from an electroplating wastewater (Hg(II) = 2 mg/L, Cr(III) = 3 mg/L, Ni(II) = 2 mg/L, Cu(II) = 1 mg/L, and Cd(II) = 3 mg/L) with a Hg(II) removal yield exceeding 99%.

## 3. Conclusions

This review shows that the topic of Hg(II) adsorption from aqueous solutions is worth to be investigated, though there is a polarization in the Institutions-Countries devoted to it, since as Table 6 showed, most of the manuscripts published during 2019 come from Mid-East and Asia (no one from Japan).

**Table 6.** Distribution by countries of manuscripts published in 2019 about Hg(II) adsorption.

| Country | Number of Manuscripts | % Over Total |
|---|---|---|
| China | 60 | 51.0 |
| India | 12 | 10.3 |
| Iran | 10 | 8.5 |
| Saudi Arabia | 5.5 | 4.7 |
| Egypt | 5 | 4.3 |
| South Korea | 3 | 2.6 |
| Pakistan | 3 | 2.6 |
| Indonesia | 2 | 1.7 |
| Malaysia | 2 | 1.7 |
| South Africa | 2 | 1.7 |
| Others (12) | 1 each | **10.3** |
| Oman | 0.5 | **0.4** |
| Total | 117 | **99.8** |

In the continuous effort to develop new adsorbents, scientists are creating a number of materials which apparently presented very different Hg(II) capacities, as the summary presented in Table 7 allows to see. It can be seen here, that Hg(II) uptake vary from very few mg/g to thousands levels. In addition, many of the published papers (near 45% over the total) forget, or do not consider, that the adsorption process must be followed by a desorption step, thus, the real usefulness of these adsorbents is not completely investigated. Moreover, in the case of manuscripts which presented data about the continuous usage (cycles) of the given adsorbents (a remarkable point seeing the previous percentage), it is shown that in the majority of the cases, and against it is stated by the respective authors, the adsorbent lost a significant capacity of its Hg(II) uptake with respect to the initial one.

**Table 7.** Selected Hg(II) uptakes.

| Hg(II) Uptake (mmol/g) | Material Type and Reference | Hg(II) Uptake (mmol/g) | Material Type and Reference |
|---|---|---|---|
| 13.27 | Alkali-modified fly ash [87] | 1.22 | Funcionalized magnetic nanocomposites [27] |
| 8.43 | Azine-linked [54] | 0.83 | Natural rubber biocomposite [97] |
| 7.95 | Funcionalized carboxymethyl chitosan composite [116] | 0.39 | Bio-apatite materials [9] |
| 4.60 | Cysteine modified cellulose nanocrystal [52] | 0.05 | Funcionalized carbon [56] |
| 3.34 | Ui-66-$NH_2$ post-funcionalization [24] | 0.03 | Magnetic hydrogel of lignin [74] |

The distribution is based in the Institution-Country of the corresponding author e-mail. Others included: Brazil, Colombia, France, Iraq, Kazakhstan, Mexico, New Zealand, Philippines, Portugal, Russia, Spain, United Arab Emirates.

Practically all of the manuscripts considered the pH, of the Hg(II)-bearing aqueous solution, as one of the variables to be investigated, and most of the manuscripts considered the 1–8 pH range for their investigations. However, and according with the stability diagrams generated from the authors of this review (Figure 6) by MEDUSA Program [121], in nitrate medium and from approximately at a pH of 4, the predominant species is Hg(OH)$_2$ which does not solubilize in the medium; in the case of chloride medium this pH value shifted until the value of 5.5. Thus, how can the results be explained?

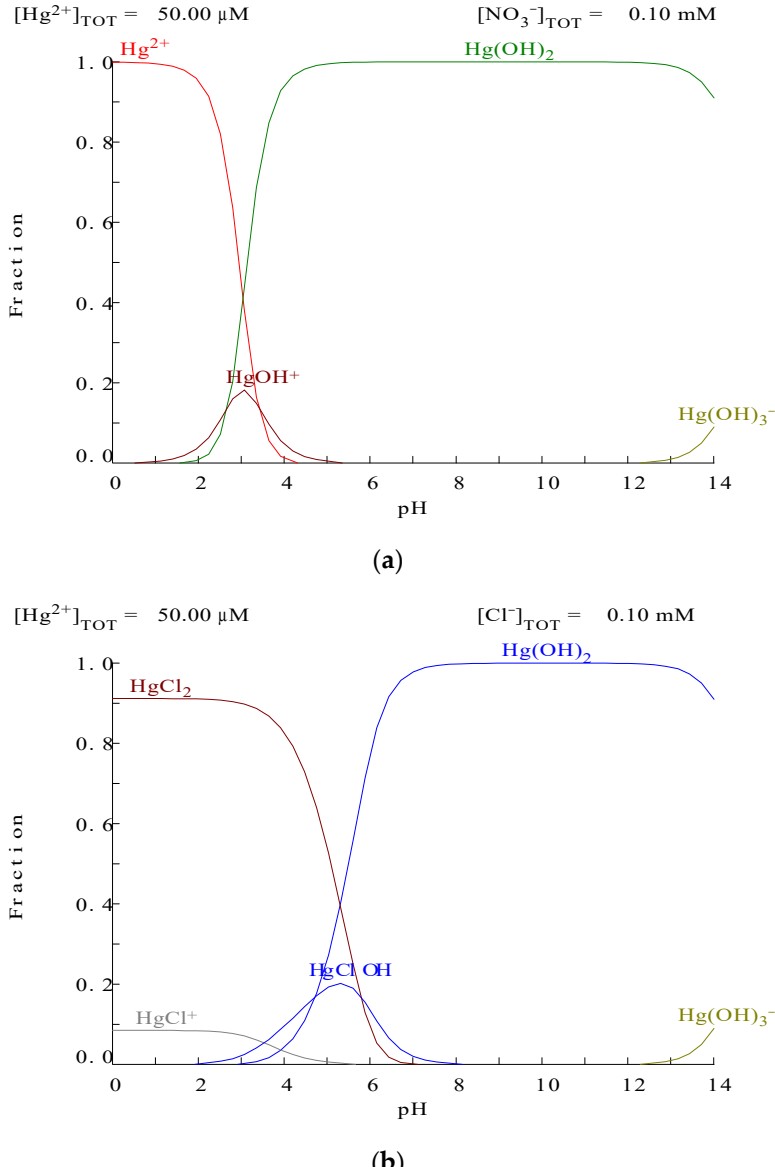

**Figure 6.** (**a**) Hg(II) species distribution against pH in nitrate medium. (**b**) Hg(II) species distribution against pH in a stoichiometric 1:2 (Hg$^{2+}$:Cl$^-$) medium.

In the case of the chloride medium, and in excess of chloride ions with respect to the Hg(II) stoichiometric concentrations, the anionic species HgCl$_3^-$ and HgCl$_4^{2-}$ become predominant at pH values lower than 6.

Of the experimental variables (batch operation) investigated in these works, there is one that it is never considered by the respective authors, and that is the influence of the stirring speed on Hg(II) adsorption. This, more than often neglected investigated variable, may have the same importance in the adsorption process as e.g., the pH of the solution, since with the correct stirring speed the contact between the adsorbent and the bulk aqueous phase is the best, resulting in a minimization of the aqueous boundary layer which maximizes metal uptake. Since it can not be given a generalization of the influence of this variable on solutes uptake, this must be experimentally proven.

Lastly, in all the manuscripts, the authors apparently have not a view of the overall problem of Hg(II) contamination in liquid effluents, that is, they do not give a solution, or simply a comment of what to do with the mercury(II) presented in the eluate; they change Hg(II) from a medium (feed solution) to another medium (eluate), nothing else.

Nevertheless, the interest to find suitable adsorbents for Hg(II) continues, and it can be said that the Hg(II)-adsorption story has not ended.

**Author Contributions:** Conceptualization, F.J.A.; methodology, F.J.A. and F.A.L.; formal analysis, F.J.A. and F.A.L.; investigation, F.J.A.; resources, F.A.L.; writing—original draft preparation, F.J.A.; writing—review and editing, F.J.A. and F.A.L. All authors have read and agreed to the published version of the manuscript.

**Funding:** This research received no external funding.

**Acknowledgments:** We acknowledge support of the publication fee by the CSIC Open Access Publication Support Initiative through its Unit of Information Resources for Research (URICI).

**Conflicts of Interest:** The authors declare no conflict of interest.

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
