# Peer review of "Adsorption Processing for the Removal of Toxic Hg(II) from Liquid Effluents: Advances in the 2019 Year"

_metals, doi:10.3390/met10030412_

Round 1

Reviewer 1 Report

It is a highly impressive compilation of the works done on the field of Hg removal. For those related can use this manuscript for understanding the global trend and the issues that are still challenging.

However, there are some points, which are better modified for improving the manuscript's impact.

  1. Using mol/kg or mmol/g instead of mg/g. This will help in understanding the ion-exchange/coordination/physisorption capacity of the materials in a glance.
  2. Table 6: In addition to the number, it would be nice to mention the major type of materials studied. MOFs, zeolites, metal oxides, or so. 
  3. Table 7: please add the material type and the condition of the highest capacity so that it becomes easier to understand what group of materials are superior for what reason. Sometimes the capacity is low only because the study was carried out taking trace concentration solution. This does not mean that the material cannot adsorb more.

Author Response

Reviewer #1

We appreciate Reviewer 1's comments and indicate our responses

  1. Using mol/kg or mmol/g instead of mg/g. This will help in understanding the ion-exchange/coordination/physisorption capacity of the materials in a glance.

Adsorption expression units have been modified throughout the text (mg / g has been modified by mmol/g)

  1. Table 6: In addition to the number, it would be nice to mention the major type of materials studied. MOFs, zeolites, metal oxides, or so. 

We believe that it is very difficult to modify Table 6 since it is impossible to summarize in a single table the data that the reviewer comments. The table is only intended to show that Hg elimination is a global problem, which worries scientists in many countries.

  1. Table 7: please add the material type and the condition of the highest capacity so that it becomes easier to understand what group of materials are superior for what reason. Sometimes the capacity is low only because the study was carried out taking trace concentration solution. This does not mean that the material cannot adsorb more.

Table 7 has been modified following the reviewer's comments

Reviewer 2 Report

The review paper titled ”Adsorption processing for the removal of toxic Hg(II) from liquid effluents: advances in the 2019 year” is a useful compilation of the recent progress achieved with advanced functional materials, however its technical quality could be improved before publication.

A major issue with the paper is that it often cites adsorption data by simply giving the removal efficiency as a percent (%) value of the original metal ion concentration. These data are meaningless without reporting i) the initial metal ion concentration together with ii) the concentration of the adsorbent in the final mixture. It is evident that the percent removal of the metal ion is directly dependent both on the concentration of the metal ion and on the concentration of the adsorbent. To be precise, the percent removal depends on the absolute concentrations and not even on the ratios of the reactants. Thus, the convenient and scientifically correct approach to benchmark adsorbents is to report the maximum adsorptive capacity together with the appropriate equilibrium constant (e.g. Langmuir) for each material.

Another major point for improvement is to exclude such papers from the review that are clearly irrational, e.g. mentioning Co(IV) or Cr(II) species, or adsorption capacities as high as 2000 mg/g, or quantifying Hg(II) adsorption under alkaline conditions, etc. These papers are evidently erroneous, and therefore, should NOT be highlighted. (And this Reviewer does acknowledge that these papers are scientifically erroneous.)  Another approach would be to critically review these erroneous papers. However, the few comments given by the authors of the present review are far too subjective and incoherent to make an impact in their present form. This is a problem, because these comments are technically correct, useful and could form the basis of a high impact critical review. Please amend.

Minor points:

Gelatin is known to have a have affinity towards Hg(II), and selective adsorbents were made from it in 2019, e.g. in DOI: 10.1021/acsanm.9b01903. Please review.

Page 3: “Comment from the authors of this review: as the readers aware, this Langmuir equation is different that shown in eq. (1), the reason is that there are up to four Langmuir solutions to explain solutes adsorption.” This comment is misleading. There is only one Langmuir model, that can be expressed in several, but completely equivalent mathematical forms.

Page 5, line 194: Please delete the subjective tone from this comment.

Page 12, line 485: Please delete the subjective tone from this comment.

Page 27, line 1081: Please delete the subjective tone from this comment.

The Hg(II) adsorption capacities in Table 7 higher than 1500 mg / g are irrational.

This Reviewer would like to highlight that it is a strength of the present review paper taking into account the speciation of aqueous Hg(II) for explaining adsorption data. Unfortunately, Fig. 6 is not displayed correctly in the ”for review” version of the manuscript.

Author Response

We appreciate Reviewer 2's comments and indicate our responses

  1. A major issue with the paper is that it often cites adsorption data by simply giving the removal efficiency as a percent (%) value of the original metal ion concentration. These data are meaningless without reporting i) the initial metal ion concentration together with ii) the concentration of the adsorbent in the final mixture. It is evident that the percent removal of the metal ion is directly dependent both on the concentration of the metal ion and on the concentration of the adsorbent. To be precise, the percent removal depends on the absolute concentrations and not even on the ratios of the reactants. Thus, the convenient and scientifically correct approach to benchmark adsorbents is to report the maximum adsorptive capacity together with the appropriate equilibrium constant (e.g. Langmuir) for each material.

We share the reviewer's opinion. However, in a review work like this we have preferred to reflect the data of the authors of the scientific papers studied, without modifying or treating them, among other things, because it is not possible to do so in all papers. For this reason, we have limited ourselves to expressing the elimination percentages as they appear in the papers studied.

  1. Another major point for improvement is to exclude such papers from the review that are clearly irrational, e.g. mentioning Co(IV) or Cr(II) species, or adsorption capacities as high as 2000 mg/g, or quantifying Hg(II) adsorption under alkaline conditions, etc. These papers are evidently erroneous, and therefore, should NOT be highlighted. (And this Reviewer does acknowledge that these papers are scientifically erroneous.) Another approach would be to critically review these erroneous papers. However, the few comments given by the authors of the present review are far too subjective and incoherent to make an impact in their present form. This is a problem, because these comments are technically correct, useful and could form the basis of a high impact critical review. Please amend.

We understand the reviewer's point of view. However, we believe that providing comparative data on the adsorption of other metals can serve to provide more complete information to the reader. We have only done this in a couple of cases and we have never commented on a paper that only studied the adsorption of these metals (eg Co,Cd or Cr). In some of the papers studied this comparative information was offered and we thought it convenient to reflect it in our manuscript .

  1. Gelatin is known to have a have affinity towards Hg(II), and selective adsorbents were made from it in 2019, e.g. in DOI: 10.1021/acsanm.9b01903. Please review.

The paper proposed by the reviewer has been included in the text and in the list of references.

  1. Page 3: “Comment from the authors of this review: as the readers aware, this Langmuir equation is different that shown in eq. (1), the reason is that there are up to four Langmuir solutions to explain solutes adsorption.” This comment is misleading. There is only one Langmuir model, that can be expressed in several, but completely equivalent mathematical forms.

We agree with the reviewer's comment. However, we have indicated the Langmuir equation that appears in the corresponding paper. We cannot modify the equations that the authors have used, since that would change the results that they have published (after peer review processes).

  1. Page 5, line 194: Please delete the subjective tone from this comment.

The comment has been removed

  1. Page 12, line 485: Please delete the subjective tone from this comment.

The comment has been removed

  1. Page 27, line 1081: Please delete the subjective tone from this comment.

The comment has been removed

  1. The Hg(II) adsorption capacities in Table 7 higher than 1500 mg / g are irrational.

These are the data that appear in the referenced papers

  1. This Reviewer would like to highlight that it is a strength of the present review paper taking into account the speciation of aqueous Hg(II) for explaining adsorption data. Unfortunately, Fig. 6 is not displayed correctly in the ”for review” version of the manuscript.

We appreciate the reviewer's comment. In the original manuscript, Figure 6 is clearly shown.